# TYPYBENCH: Evaluating LLM Type Inference for Untyped Python Repositories

Honghua Dong [* 1 2]   Jiacheng Yang [* 1 2]   Xun Deng [* 1]   Yuhe Jiang [* 1 2]   Gennady Pekhimenko [1 2 3]   Fan Long [1]
Xujie Si [1 2 3]

## Abstract

Type inference for dynamic languages like Python is a persistent challenge in software engineering. While large language models (LLMs) have shown promise in code understanding, their type inference capabilities remain underexplored. We introduce TYPYBENCH, a benchmark designed to evaluate LLMs' type inference across entire Python repositories. TYPYBENCH features two novel metrics: TYPESIM, which captures nuanced semantic relationships between predicted and ground truth types, and TYPECHECK, which assesses type consistency across codebases. Our evaluation of various LLMs on a curated dataset of 50 high-quality Python repositories reveals that, although LLMs achieve decent TYPESIM scores, they struggle with complex nested types and exhibit significant type consistency errors. These findings suggest that future research should shift focus from improving type similarity to addressing repository-level consistency. TYPYBENCH provides a foundation for this new direction, offering insights into model performance across different type complexities and usage contexts. Our code and data are available at https://github.com/typybench/typybench.

## 1. Introduction

Type inference, the ability to automatically deduce the types of variables and expressions in a program, has been a long-standing challenge in programming language research (Raychev et al., 2015; Hellendoorn et al., 2018). In dynamically-typed languages like Python, where explicit type annotations are optional, type inference involves analyzing code to determine appropriate type annotations that could have been written by developers. This capability has become increasingly important as codebases grow in size and complexity.

The significance of type information in modern software development cannot be overstated. Type annotations serve multiple crucial purposes: (1) they enhance code clarity by making developers' intentions explicit, (2) prevent type-related errors through early detection, (3) enable rich IDE features like autocompletion, and (4) facilitate maintenance and refactoring operations. The introduction of type hints through PEP 484[1] marked a pivotal moment for Python, acknowledging the growing importance of static typing in large-scale software development.

While the benefits of type annotations are clear, manually adding them to existing codebases is time-consuming and error-prone. This challenge has sparked interest in developing automatic type inference tools like Mypy (Lehtosalo et al.), Pyright (Microsoft) and MonkeyType (Instagram), as well as learning-based algorithms (Wei et al., 2020; Allamanis et al., 2020). Moreover, recent advances in large language models (LLMs) have shown promising results in code understanding tasks (Brown et al., 2020; Achiam et al., 2023; Chen et al., 2021), and in type inference tasks (Wei et al., 2023; Peng et al., 2023) with better performance than previous methods (Shivarpatna Venkatesh et al., 2024). Such tools can significantly reduce developer effort while improving code quality and maintainability.

Despite these advances, current evaluation benchmarks and approaches (Mir et al., 2021; Allamanis et al., 2020; Shivarpatna Venkatesh et al., 2024) for type inference methods face significant limitations. Traditional evaluation metrics rely heavily on exact matching (or up to parametric type (Allamanis et al., 2020)), which fails to capture important semantic and functional relationships between types – for instance, the functional similarity between `List` and `Sequence`, where developers may use them interchangeably. Furthermore, existing benchmarks often evaluate type inference in isolation, focusing on individual functions or files rather than considering type consistency across entire codebases. This disconnect between local correctness and global coherence makes it challenging to reliably assess the real-world effectiveness of type inference methods.

---

* Equal contribution, alphabetically ordered. [1]University of Toronto [2]Vector Institute [3]CIFAR AI Chair. Correspondence to: Xujie Si <six@cs.toronto.edu>.

*Proceedings of the 42nd International Conference on Machine Learning*, Vancouver, Canada. PMLR 267, 2025. Copyright 2025 by the author(s).

---

[1]https://peps.python.org/pep-0484/

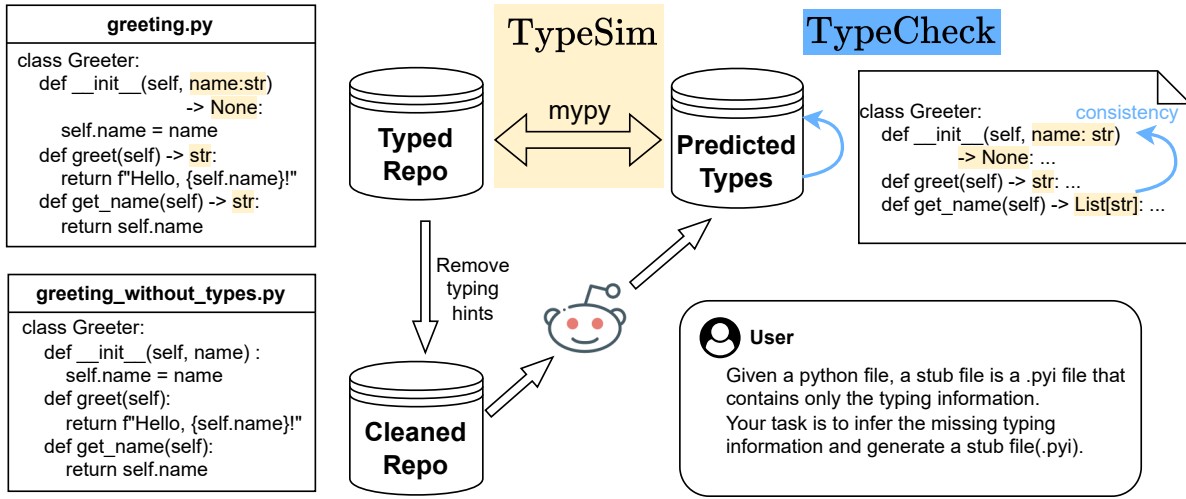

Figure 1. Overview of TYPYBENCH. We collect well-typed Python repositories and remove their typing information as the inputs. The outputs predicted by type inference methods are then evaluated using TYPESIM and TYPECHECK, where TYPESIM measures the functionality similarity between predicted and human-annotated types, and TYPECHECK evaluates type consistency across entire codebases through static type checking.

To address these challenges, we introduce two novel evaluation metrics, TYPESIM and TYPECHECK, as illustrated in Figure 1. TYPESIM measures the functional similarity between predicted and human-annotated types by incorporating both structural relationships and type hierarchies, providing a more nuanced evaluation than exact matching. Complementarily, TYPECHECK measures type consistency across entire codebases via static type checking[2], verifying that inferred types integrate coherently.

We then present TYPYBENCH, a benchmark of 50 high-quality Python repositories selected based on their type coverage, complexity, and domain diversity. Unlike existing benchmarks (Pradel et al., 2020; Mir et al., 2021), TYPYBENCH emphasizes repository-level evaluation, allowing the assessment of both local type accuracy and global type consistency.

Our extensive evaluation of state-of-the-art LLMs on TYPYBENCH reveals key insights. While modern LLMs achieve decent TYPESIM scores (up to 0.80), most of them struggle with TYPECHECK, highlighting a gap between local type accuracy and global consistency compared to human annotations. The main contributions of this paper are:

- Two novel metrics for evaluating type inference: TYPESIM for measuring semantic similarity between types, and TYPECHECK for assessing repository-wide type consistency.
- TYPYBENCH: A comprehensive benchmark of 50 high-

quality Python repositories, designed to evaluate both local and global aspects of type inference.

- An extensive empirical study of state-of-the-art LLMs on type inference, revealing critical gaps between type prediction accuracy and consistency.

These contributions establish a foundation for future research in automated type inference, providing both the tools and insights needed to develop more effective type inference systems.

## 2. Related Work

### 2.1. Type Inference Methods

**Conventional Methods.** Traditional type inference relies on static analysis and runtime tracing, as implemented in tools like Mypy (Lehtosalo et al.), Pyright (Microsoft), and MonkeyType (Instagram). These approaches offer high precision but are limited in coverage and require explicit type annotations or runtime information.

**Learning-based Methods.** Early learning approaches like JSNice (Raychev et al., 2015) pioneered using probabilistic models to learn from existing codebases. This direction evolved to leverage various program representations, from natural language information (Malik et al., 2019) to graph structures (Hellendoorn et al., 2018; Wei et al., 2020; Allamanis et al., 2020; Cassano et al., 2023), though struggling to handle complex type structures and rare types.

**LLM-based Methods.** Recent work has shown that large language models can match or exceed traditional approaches

---

[2]We use the number of Mypy check errors to estimate the consistency of types.

in type inference tasks (Jesse et al., 2021; Wei et al., 2023; Peng et al., 2023). These methods benefit from pre-training on large code corpora and can leverage natural language understanding for improved type prediction.

## 2.2. Type Inference Benchmarks

Previous type inference benchmarks (Pradel et al., 2020; Mir et al., 2021; Allamanis et al., 2020) primarily relied on exact match accuracy for evaluation, with Typilus (Allamanis et al., 2020) introducing a relaxed "Match up to Parametric Type" metric that compares only the outermost type constructors. However, these metrics still fall short of capturing full semantic similarity between type annotations. Our benchmark advances this by (1) introducing semantic similarity metrics that better capture the hierarchical and structural relationships between types, and (2) evaluating practical usability through type checking. By requiring predictions in the form of PEP 484 stub files (.pyi), we enable direct validation using production-grade type checkers, providing a more realistic assessment of type inference quality.

## 2.3. Other Coding Benchmarks

The evolution of code-related benchmarks reflects a progression from isolated to context-dependent evaluations:

**Function-level Benchmarks.** Traditional benchmarks focused on self-contained programming tasks (Chen et al., 2021; Jain et al., 2024; Zhuo et al., 2024), evaluating specific capabilities like code generation and problem-solving.

**Repository-level Benchmarks.** Recent work has shifted toward repository-scale assessment, with each benchmark evaluating distinct aspects of code understanding: RepoBench (Liu et al., 2024b) focuses on code completion, SWE-bench (Jimenez et al., 2024) tests bug fixing capabilities, and RepoTransBench (Wang et al., 2024) evaluates cross-language translation. Our work contributes to this ecosystem by examining models' ability to perform consistent type inference across entire repositories, adding another crucial dimension to repository-level model evaluation.

## 3. Background

This section introduces key concepts in Python's type system and type inference that are fundamental to our work.

## 3.1. Gradual Typing in Python

Python supports gradual typing, allowing developers to incrementally add type annotations while maintaining compatibility with untyped code. Introduced in PEP 484, type hints enable specifying types for function parameters, return values, and variables:

```
1  def greet(name: str) -> str:
2      return f"Hello, {name}!"
```

These optional annotations do not affect runtime behavior, serving primarily as documentation and enabling static analysis tools to catch type-related errors before execution.

## 3.2. Type Inference

Type inference is the process of automatically deducing appropriate type annotations for variables and expressions in a program. In our context, given a Python repository without type annotations, the goal is to infer types that could have been written by developers:

```
1  # Original untyped        1  # After type inference
     code                     2  def greet(name: str)
2  def greet(name):                -> str:
3      return f"Hello,         3      return f"Hello,
         {name}!"                      {name}!"
```

Here, the return type is `str` based on the returned value. The parameter `name` is likely to be `str` based on the semantic information, since the name of the variable is `name` and the function is `greet`.

## 3.3. Type Stub Files

Python uses `.pyi` stub files to separate type information from implementation. These files contain only function signatures and type definitions:

```
1  # greetings.pyi
2  def greet(name: str) -> str: ...
```

Stub files enable type checking without modifying source files and are commonly used in library distributions to provide type information.

## 3.4. Static Type Checking

Static type checking verifies type consistency before program execution. Tools like mypy[3] analyze code to detect potential type errors, which include but are not limited to: (1) verifying type compatibility in assignments and function calls, (2) checking subtype relationships (e.g., `List[int]` is a subtype of `Sequence[int]`), (3) ensuring consistent usage of types across modules. In our example, mypy would detect errors like:

```
1  msg = greet("TypyBench") # Pass, "TypyBench" is str
2  msg = greet([1, 2, 3]) # Error: Expected str, got
     list[int]
```

Mypy performs this analysis by constructing a type dependency graph and propagating type constraints through the program, identifying violations of type rules defined in PEP 484 and related specifications.

---

[3]https://mypy.readthedocs.io/

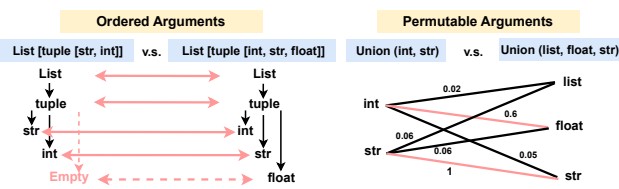

*Figure 2.* Examples of type similarity computation. Left: List-wise comparison for generic types, where arguments are compared in order. Right: Set-wise comparison for Union types, where an optimal matching is computed between members.

# 4. Metric Design

To effectively evaluate type inference systems, we introduce two complementary metrics: TYPESIM, which measures type prediction quality, and TYPECHECK, which assesses type coherence across codebases.

## 4.1. Type Similarity

Traditional evaluation methods for type inference rely on exact matching, which fails to capture the nuanced relationships between types. For example, Sequence[int] and Iterable[int] share most functionality but would receive a score of 0 under exact matching, even though both support iteration operations. Similarly, int and float would be considered completely different despite sharing most arithmetic operations. To solve this issue, we propose TYPESIM (Algorithm 1), a continuous similarity metric that considers both functional similarity and structural relationships.

### 4.1.1. BASE TYPE SIMILARITY

For non-generic types, we compute similarity based on their supported operations and methods. Given two types $t$ and $t'$, their similarity is:

$$s(t, t') = \frac{|attrs(t) \cap attrs(t')|}{|attrs(t) \cup attrs(t')|}$$

where $attrs(t)$ represents the set of methods and operations supported by type $t$, excluding those common to all types that inherit from object (like __str__ or __init__). We then use the Jaccard index to measure functional similarity, where two types are similar if they share most of the same methods and operations. This approach captures the intuition that types supporting similar operations should be considered similar.

Consider Python's collection types hierarchy as an example:

- Iterable provides __iter__ for iteration, enabling for loops.

- Sequence adds index, count, and length operations to Iterable, supporting indexed access.

- List adds mutable operations like append, extend, pop to Sequence, enabling list modification.

This leads to meaningful similarities: $s(\text{Iterable}, \text{Sequence}) = 0.92$ as they share core iteration functionality, while $s(\text{Sequence}, \text{List}) = 0.7$ reflects their additional differences in mutability. Similarly, $s(\text{int}, \text{float}) = 0.6$ captures their shared arithmetic operations, while $s(\text{int}, \text{str}) = 0.06$ reflects their fundamental differences. See Appendix B for the TYPESIM between builtin types.

### 4.1.2. STRUCTURAL SIMILARITY

**ListCompare.** For generic types (e.g., List[int], Dict[str, int]), we compute similarity recursively through their type tree structure, as shown in Figure 2. This allows us to handle nested types of arbitrary depth while considering both the container types and their type arguments.

As shown on the left of Figure 2, for non-union types $T$ and $T'$, their similarity is:

$$S(T, T') = \frac{1}{2}(s(root, root') + S_{list}(args(T), args(T'))),$$

where $s(root, root')$ is the base type similarity, and $S_{list}$ (Algorithm 2) compares type arguments in order:

$$S_{list}(L, L') = \frac{\sum_{i \leq \min(|L|, |L'|)} S(L_i, L'_i)}{\max(|L|, |L'|)}.$$

For example, comparing List[int] with Sequence[float]:

- Base similarity: $s(\text{List}, \text{Sequence}) = 0.7$ (shared sequence operations)

- Argument similarity: $S(\text{int}, \text{float}) = 0.6$ (shared numeric operations)

- Overall: $S = \frac{1}{2}(0.7 + 0.6) = 0.65$

We choose to average the similarity of the root and the arguments instead of multiplying them to give more weight to root types. This design aligns with common development practices where developers often annotate only the root type (e.g., List) without specifying arguments, especially during initial typing efforts. For example, when computing

---

**Algorithm 1** TYPESIM

---

   **Input:** types $T, T'$
   **if** At least one of $T, T'$ is Union **then**
      **Return: SetCompare**$(as\_set(T), as\_set(T'))$
   **end if**
   $score = s(T.root, T'.root)$
   **if** Both $T, T'$ have arguments **then**
      $score = \frac{1}{2}(score + \textbf{ListCompare}(T.args, T'.args))$
   **else if** One of $T, T'$ has arguments **then**
      $score = \frac{score}{2}$
   **end if**
   **Return:** $score$

---

**Algorithm 2** ListCompare

---

   **Input:** type lists $L, L'$
   // List-wise comparison for generic type arguments
   $S = 0$
   **for** $i = 1$ **to** $\min(|L|, |L'|)$ **do**
      $S = S + \textbf{GetTypeSimilarity}(L_i, L'_i)$
   **end for**
   **Return:** $\frac{S}{\max(|L|, |L'|)}$

---

$S(\texttt{List}, \texttt{List[int]})$, while the argument similarity is 0 due to the missing argument, the base similarity is 1 for matching root types. Averaging yields 0.5, acknowledging the partial correctness of the annotation, whereas multiplication would give 0, completely penalizing this common and often acceptable practice in gradual typing.

**SetCompare.** As shown on the right of Figure 2, we treat union types (e.g., $\texttt{Union[int, str]}$) as sets of possible types and compute an optimal matching between their members. This approach accounts for unordered union members and allows partial matches.

Given types $T$ and $T'$, where at least one is a union:

$$S(T, T') = S_{set}(as\_set(T), as\_set(T')),$$

where $as\_set(T)$ converts a type to its member set:

$$as\_set(T) = \begin{cases} \{T\} & \text{if T is not Union.} \\ \{T.args\} & \text{if T is Union.} \end{cases}$$

$S_{set}$ (Algorithm 3) finds the optimal matching between members:

$$S_{set}(A, B) = \frac{\sum_{(i,j) \in M} S(A_i, B_j)}{\max(|A|, |B|)}.$$

### 4.2. Type Consistency

While TYPESIM measures how close the predictions are to human annotations, TYPECHECK evaluates whether the

---

**Algorithm 3** SetCompare

---

   **Input:** type sets $A, B$
   // Set-wise comparison for Union types
   **for** $i = 1$ **to** $|A|$ **do**
      **for** $j = 1$ **to** $|B|$ **do**
         $c_{ij} = \textbf{GetTypeSimilarity}(A_i, B_j)$
      **end for**
   **end for**
   Find optimal matching $M$ using cost matrix $C = [c_{ij}]$
   $S = \sum_{(i,j) \in M} c_{ij}$
   **Return:** $\frac{S}{\max(|A|, |B|)}$

---

predicted types form a coherent system across the codebase. Since type annotations serve not only as documentation but also as a mechanism for early error detection and code maintenance, it is crucial that predicted types work together consistently across the entire codebase.

We use the number of mypy errors as a proxy for type consistency because these errors directly reflect what developers would need to fix before the type annotations become practically useful. For example, if a function is predicted to return $\texttt{List[int]}$ but is used in a context expecting $\texttt{List[str]}$, this inconsistency would prevent effective static type checking and IDE support – two key benefits of type annotations. Specifically, we focus on meaningful type errors that affect code correctness, such as incompatible return types and invalid argument types. These errors indicate real issues that would hinder code maintenance and refactoring. A complete list of counted error types is provided in Appendix C.2.

## 5. Dataset Curation

We curate a benchmark dataset containing 50 popular Python repositories from GitHub and PyPI to evaluate type inference capabilities. The dataset construction involves two key steps: repository selection and cleaning.

### 5.1. Repository Selection

We select candidate repositories from GitHub's trending repositories and frequently downloaded PyPI Packages. In the initial filtering stage, we enforce the constraints of a maximum of 1.5M tokens, a minimum of 30 Python files, at least 50% typed functions, and valid mypy configurations to ensure the suitability of type inference evaluation. We then define a quality score as below and rank the candidate repositories by their score:

$$S = \alpha S_{\text{coverage}} + \beta S_{\text{popularity}} + \gamma S_{\text{complexity}},$$

where $S_{\text{coverage}}$ measures the percentage of typed functions, $S_{\text{popularity}}$ is calculated from the number of GitHub stars

**Algorithm 4** Type Removal Process

  **Input:** Repository $R$ with typed Python files
  **Output:** Repository $R'$ with types removed
  **for** each Python file $f \in R$ **do**
    Parse AST of $f$ to locate type annotations
    **for** each function/variable declaration $d$ **do**
      **if** $d$ has type annotation $t$ **then**
        Remove $t$ from $d$ in $f$ {e.g., def foo(x: int) → def foo(x)}
      **end if**
      **if** the docstring $c$ contains type hint for the function arguments or returns (or variable annotations) **then**
        Remove the type hint
      **end if**
    **end for**
    Save modified file to $R'$
  **end for**
  **Return:** $R'$

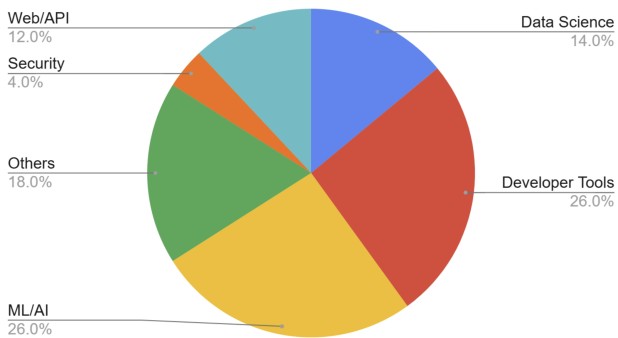

*Figure 3.* Distribution of repository categories in our benchmark dataset, showing coverage across major Python application domains.

### 5.3. Benchmark Statistics

To facilitate the potential evaluation of learning-based methods, we split the 50 repositories into Train/Validation/Test splits, with 20/10/20 repositories correspondingly. The 20 test repositories are selected and further split into two sets with 10 repositories each based on the date created to estimate the level of data contamination.

Table 1 summarizes the number of tokens, functions, and variables to be inferred for different splits. We also depict the diversity of TYPYBENCH in Figure 3 by classifying the repositories into domains spanning Developer Tools, ML/AI, Web/API, and Security. For more comprehensive details about each repository, please refer to Appendix A.

*Table 1.* The total number of tokens, functions, and variables to be inferred for different splits.

|            | # Repos | # Tokens | # Functions | # Cases |
|------------|---------|----------|-------------|---------|
| Train      | 20      | 8403760  | 31161       | 59966   |
| Validation | 10      | 4037666  | 13988       | 20177   |
| Test       | 20      | 4983025  | 25101       | 46166   |

and PyPI downloads, and $S_{\text{complexity}}$ considers the depth and variety of type annotations. The top 50 repositories with the highest candidate scores are selected into the dataset.

### 5.2. Repository Cleaning

To ensure the best quality of type inference evaluation, we remove non-Python files, testing files, and irrelevant files, to only keep the Python files in the source folder containing the main functionality. To ensure the quality of type annotations, we run mypy check on the original repository, and manually resolve errors that stop the check from running.

After initial cleaning, as illustrated in Figure 1, we remove type annotations using scripts while preserving code functionality to create input and ground truth pairs for evaluation. The type removal algorithm handles function signatures, and variable declarations as shown in Algorithm 4.

To maximize consistency and similarity with the original repository, every single part of each Python program is kept unchanged except for the type annotation. We rebuild each processed repository to ensure that the modified code remains syntactically valid and the runtime behaviors are preserved.

## 6. Experiments

In this section, we conduct comprehensive experiments to answer the following key questions:

- **Model Readiness:** How well do current LLMs perform on type inference? Are SOTA models ready for production use on untyped repositories? What are their key limitations? Does a longer context length help mitigate the limitations?

- **Metric Effectiveness:** How do our proposed TYPE-SIM and TYPECHECK metrics compare to traditional exact matching? What additional insights do they provide?

- **Performance Factors:** How do factors like type complexity and repository age affect model performance? What do these patterns reveal about LLMs' type inference capabilities?

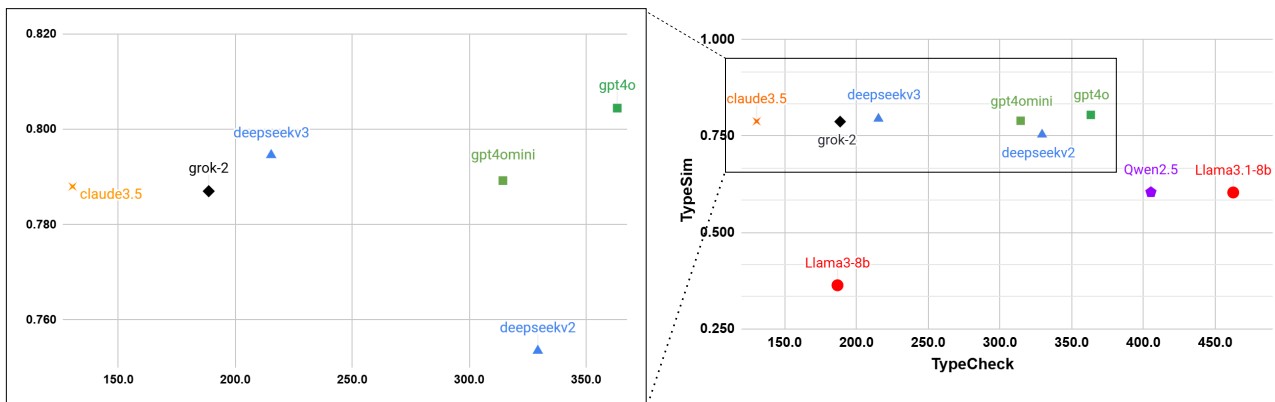

*Figure 4.* Comparison of TYPESIM and TYPECHECK metrics across models. SOTA models (CLAUDE-3.5-SONNET, DEEPSEEK-V3, GPT-4O, GPT-4O-MINI, and GROK-2) achieve similar high TYPESIM scores while varying in TYPECHECK, showing TYPECHECK's additional discriminative power. Small local-hosted models (LLAMA-3-8B, LLAMA-3.1-8B, QWEN-2.5-7B) perform poorly.

### 6.1. Experimental Setup

We evaluate type inference capabilities across a diverse set of SOTA LLMs, including API-accessed models and local-hosted small models. The API-based LLMs include GPT-4O, GPT-4O-MINI (Achiam et al., 2023), CLAUDE-3.5-SONNET (Anthropic, 2024), DEEPSEEK-V2.5, DEEPSEEK-V3 (Liu et al., 2024a), and GROK-2 (XAI, 2024). For local-hosted models, we evaluate popular models including LLAMA-3-8B, LLAMA-3.1-8B (Dubey et al., 2024), and QWEN-2.5-7B (Yang et al., 2024).

While our benchmark provides train/validation/test splits for future learning-based methods, we evaluate pre-trained LLMs on all repositories since fine-tuning is not performed on the training set. We separately analyze potential pre-training data contamination through temporal analysis (Section 6.3).

Since most repositories contain more number of tokens than the context length of most LLMs, LLMs are tested in a file-by-file method, where we require LLMs to infer the `.pyi` stub file from the type-removed file with the same prompt (see Figure 9 and Appendix C for more details).

### 6.2. Main Evaluation Analysis

As shown in Table 2, our analysis reveals both promising advances and concerning limitations in LLM-based type inference. While SOTA models achieve decent TYPESIM scores around 0.80, their non-negligible missing rates suggest systematic limitations in stably generating correct stub files. Moreover, as illustrated in Figure 4, the TYPECHECK metric reveals an interesting pattern in model consistency. While most models struggle with type checking, CLAUDE-3.5-SONNET is the best one to keep the type predictions consistent despite not having the highest TYPESIM. It is worth noting that CLAUDE-3.5-SONNET achieves a better

*Table 2.* Average TYPESIM and TYPECHECK scores of all repositories for various models. CLAUDE-3.5-SONNET shows the best TYPECHECK score while top models share similar TYPESIM scores. The ground truth has a TYPECHECK score of 141.8.

| MODEL | TYPECHECK ↓ | TYPESIM ↑ | TYPESIM WO MISSING ↑ | MISSING RATE ↓ |
|---|---|---|---|---|
| LLAMA-3-8B | 187.5 | 0.363 | 0.731 | 0.508 |
| LLAMA-3.1-8B | 465.7 | 0.603 | 0.804 | 0.261 |
| QWEN-2.5-7B | 411.0 | 0.604 | 0.787 | 0.238 |
| GPT-4O | 366.0 | **0.804** | 0.893 | **0.099** |
| GPT-4O-MINI | 310.6 | 0.789 | 0.893 | 0.116 |
| CLAUDE-3.5-SONNET | **127.1** | 0.788 | 0.893 | 0.119 |
| DEEPSEEK-V2.5 | 328.7 | 0.754 | **0.907** | 0.169 |
| DEEPSEEK-V3 | 214.6 | 0.795 | 0.897 | 0.115 |
| GROK-2 | 190.1 | 0.787 | 0.903 | 0.129 |

overall TYPECHECK score than the ground truth (141.8 errors on average), though it still performs worse than the ground truth on 29 repositories. This consistency could be particularly valuable when humans need to fix the type-checking errors manually.

### 6.3. Factors Analysis

**Impact of Type Complexity.** We first compare TYPESIM with exact match metrics for types with different depths. Table 3 reveals an increasingly widening gap at higher depths. While both metrics show declining trends, exact match scores drop more precipitously – nearly vanishing for types of depth 3 and above. In contrast, TYPESIM still captures semantically valid predictions that would be completely rejected by exact matching. This demonstrates TYPESIM's value in providing more nuanced evaluation, particularly for complex types where multiple valid type annotations may exist. As shown in Figure 5, model performance consistently slightly degrades as type complexity increases with increased variance. Even SOTA models struggle with deeper nested types (depth > 2), suggesting that

*Table 3.* Compare TYPESIM and exact match by type depth, the difference increases with depth.

| | TYPESIM ↑ | | | | | EXACT MATCH ↑ | | | | |
|---|---|---|---|---|---|---|---|---|---|---|
| MODEL | DEPTH1 | DEPTH2 | DEPTH3 | DEPTH4 | DEPTH5 | DEPTH1 | DEPTH2 | DEPTH3 | DEPTH4 | DEPTH5 |
| LLAMA-3-8B | 0.415 | 0.271 | 0.235 | 0.209 | 0.136 | 0.406 | 0.197 | 0.113 | 0.056 | 0.023 |
| LLAMA-3.1-8B | 0.635 | 0.555 | 0.499 | 0.478 | 0.454 | 0.610 | 0.427 | 0.279 | 0.201 | 0.118 |
| QWEN-2.5-7B | 0.658 | 0.525 | 0.475 | 0.443 | 0.438 | 0.640 | 0.420 | 0.277 | 0.191 | 0.113 |
| GPT-4O | 0.822 | 0.775 | 0.753 | 0.733 | 0.776 | 0.792 | 0.653 | 0.497 | 0.417 | 0.350 |
| GPT-4O-MINI | 0.813 | 0.747 | 0.703 | 0.721 | 0.705 | 0.782 | 0.616 | 0.420 | 0.370 | 0.209 |
| CLAUDE-3.5-SONNET | 0.801 | 0.768 | 0.746 | 0.749 | 0.746 | 0.769 | 0.652 | 0.517 | 0.427 | 0.303 |
| DEEPSEEK-V2.5 | 0.771 | 0.722 | 0.697 | 0.627 | 0.676 | 0.745 | 0.609 | 0.462 | 0.320 | 0.218 |
| DEEPSEEK-V3 | 0.809 | 0.769 | 0.747 | 0.702 | 0.708 | 0.774 | 0.644 | 0.483 | 0.397 | 0.295 |
| GROK-2 | 0.802 | 0.757 | 0.737 | 0.710 | 0.743 | 0.771 | 0.642 | 0.496 | 0.387 | 0.274 |
| AVERAGE | 0.725 | 0.654 | 0.621 | 0.597 | 0.598 | 0.699 | 0.540 | 0.394 | 0.307 | 0.211 |
| DIFF | | | | | | **-0.026** | **-0.114** | **-0.228** | **-0.289** | **-0.387** |

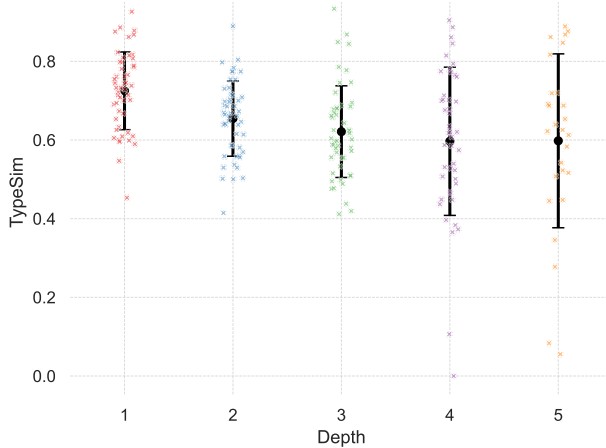

*Figure 5.* TYPESIM scores v.s. type complexity. Models achieve consistent high scores for simple types (depths 1) but show degraded performance and increased variance for more complex types (depths 2+).

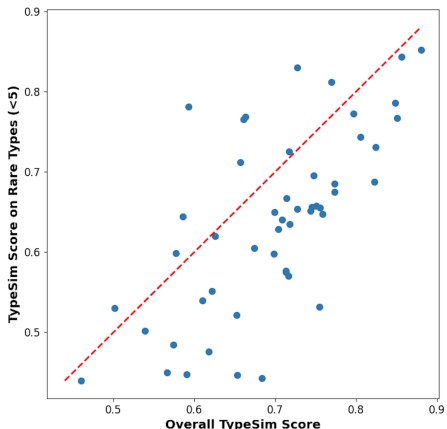

*Figure 6.* Overall TYPESIM scores v.s. TYPESIM scores on rare types. A Scatter plot of average TYPESIM score on rare types and all types for each repository. A clear drop on the score is observed on 40 out of 50 repositories (under the red line).

complex type inference remains a significant challenge in some cases.

**Impact of Type Frequency.** As pointed out in previous work (Allamanis et al., 2020), predicting rare types that are less frequent in the repository is a challenge. As shown in Table 2, we still observe a gap between the TYPESIM scores on all types and the scores on rare types, but the gap is not that large overall. However, when it comes to specific repositories, as shown in Figure 6, the drop is significant in many repositories, indicating predicting rare types is still challenging for LLMs. An interesting observation is that TYPESIM on rare types could also be higher than the overall TYPESIM in a minority of repositories.

**Impact of Repo-Level Context.** To reduce the TYPE-CHECK errors in the predicted types, the most straightfor-

ward way is to use the whole repository as the context given to LLMs. However, this approach faces two major challenges, long context length and long output length, due to the size of the whole repository. Nevertheless, we tested this approach with GPT-4O on 3 repositories with total number of tokens smaller than 64k. As shown in Table 4, the TYPECHECK errors do decrease significantly with the whole repository context, with the cost of worse TYPESIM scores, potentially due to harder to exact the correct information from the long context and harder to generate all the predictions in the correct format. This suggests that more context is helpful for enhancing the type consistency, but the long input and output challenges need to be addressed.

**Impact of Data Contamination.** As observed in previous work (Roberts et al., 2023), data contamination is a potential issue when evaluating the performance of LLMs. To verify

*Table 4.* Comparision between full repo context and single file context with GPT-4O. The TYPECHECK errors decrease significantly with the whole repository context, but the TYPESIM scores also decrease significantly on some repositories.

| MODEL | REPO | FULL REPO | | SINGLE FILE | |
|---|---|---|---|---|---|
| | | TYPESIM ↑ | TYPECHECK ↓ | TYPESIM ↑ | TYPECHECK ↓ |
| | GPTME | 0.966 | 15 | 0.877 | 84 |
| GPT-4O | PRIVATE-GPT | 0.601 | 4 | 0.855 | 17 |
| | SCREENSHOT-TO-CODE | 0.696 | 0 | 0.915 | 6 |

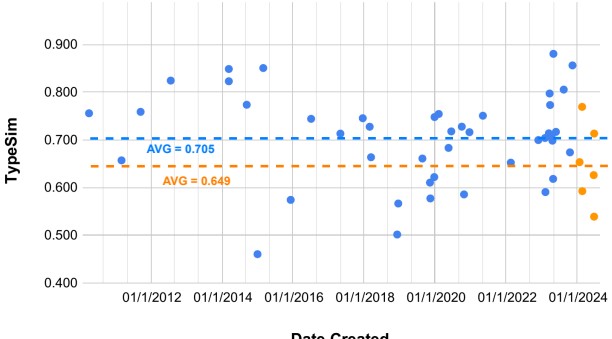

*Figure 7.* TYPESIM scores v.s. repository creation dates. Better performance observed on older repositories (pre-2024 avg: 0.705) compared to newer ones (2024+ avg: 0.649), indicating potential data contamination effects.

this, we compare the TYPESIM scores of repositories created in different years. As shown in Figure 7, the TYPESIM scores of repositories created in 2024 and after are lower than that of repositories created before 2024, suggesting potential data contamination effects. So we suggest that future work should use the test set that contains relatively recent repositories when comparing the performance of different models, and we include the results of the test set in Table 8 in Appendix D.

## 7. Conclusion

We present TYPYBENCH, a comprehensive benchmark for evaluating LLMs' Python type inference capabilities. Our evaluation reveals that while SOTA models achieve promising TYPESIM scores, they still face significant challenges: poor handling of complex nested types, and substantial type consistency errors. The proposed TYPESIM and TYPECHECK metrics provide complementary insights, with TYPESIM capturing semantically valid predictions and TYPECHECK revealing critical consistency issues. The experimental finding suggests that the focus of type inference should turn to the repo-level consistency since the similarity is already high. We further find that increased context length improves type consistency but creates challenges in handling long inputs and outputs, suggesting the need for more efficient context handling mechanisms. We hope

TYPYBENCH will facilitate progress in LLM-based type inference by providing a standardized evaluation framework and highlighting key areas for improvement.

## Impact Statement

This paper presents work whose goal is to advance the field of Machine Learning and Software Engineering. There are many potential societal consequences of our work, none which we feel must be specifically highlighted here.

## Acknowledgments

We thank Jialiang Sun for his valuable contributions to data collection and curation, Shiwen Wu for her insightful discussion, and Chris J. Maddison for his support on this project. This research project has benefitted from the Microsoft Accelerate Foundation Models Research (AFMR) grant program.

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

## A. Repository Information

The statistics of all repositories are shown in Table 5. We use the tokenizer `tiktoken_cl100k_base` to count the number of tokens.

### A.1. TypeCheck Results

Table 6 shows the results of running `mypy check` in the original repository for all repositories (except the ones are slow running mypy). We still include some repositories that have a few `mypy` errors making room for improvement.

## B. Basic Type Similarities

We illustrate the TYPESIM between builtin types in Figure 8, and list the TYPESIM scores for some similar types in Table 7.

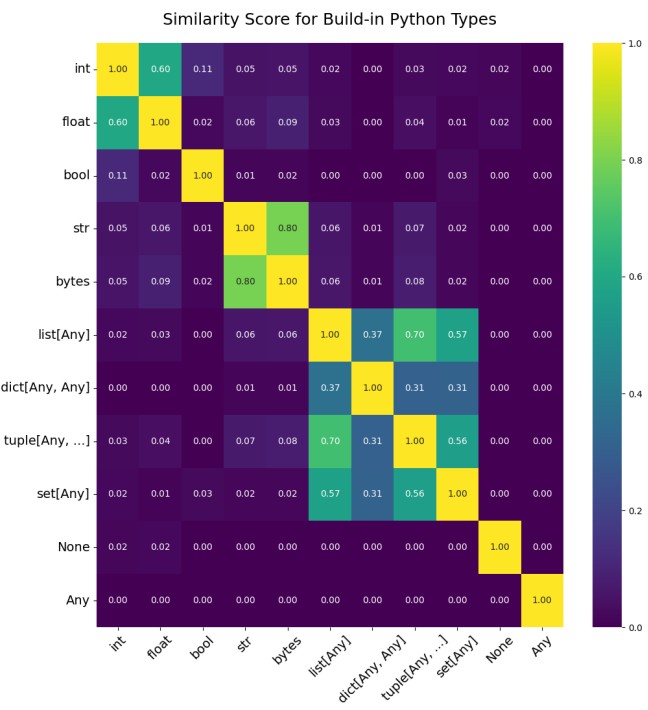

*Figure 8.* TYPESIM scores between builtin types.

## C. Experimental Settings

We use Python version `3.12` and Mypy version `1.11.1` for all our experiments. For API models with multiple versions, we use GPT-4O-2024-08-06, GPT-4O-MINI-2024-07-18, CLAUDE-3-5-SONNET-20240620, GROK-2-1212, and DEEPSEEK-V3 (12-26 VERSION).

We use APPL (Dong et al., 2024) to implement the evaluation for LLMs.

### C.1. Prompt

Figure 9 illustrates the prompt used for single-file type inference in our experiment. The prompt includes a Python file along with its corresponding .pyi file as an example. Additionally, it specifies the expected answer format and emphasizes that the generated output must be free of syntax errors.

*Table 5.* The statistics of all repositories in TYPYBENCH, including the total number of tokens, functions, variables to be inferred, the ratio of functions being annotated, the category, and the date created. The test sets are further split into two test1 and test2 based on the date created to test the data contamination issue.

| REPOSITORY | # TOKENS | # FUNC | # CASES | FUNC. ANNOT. | AVG DEPTH | CATEGORY | DATE CREATED |
|---|---|---|---|---|---|---|---|
| | | | TRAIN | | | | |
| PRE-COMMIT-HOOKS | 16268 | 105 | 810 | 1.00 | 1.57 | DEVELOPER TOOLS | 2014/03/13 |
| FLAKE8 | 34684 | 209 | 747 | 1.00 | 1.4 | DEVELOPER TOOLS | 2014/09/13 |
| VNPY | 47762 | 415 | 909 | 0.99 | 1.11 | DATA SCIENCE | 2015/03/02 |
| GHUNT | 63135 | 284 | 2185 | 0.62 | 1.91 | SECURITY | 2020/10/02 |
| SPOTIFY-DOWNLOADER | 89384 | 232 | 2143 | 0.90 | 1.32 | OTHERS | 2016/07/06 |
| DEEPFACE | 90831 | 194 | 3263 | 0.57 | 1.4 | OTHERS | 2020/02/08 |
| URLLIB3 | 98963 | 551 | 1844 | 1.00 | 1.49 | DEVELOPER TOOLS | 2011/09/18 |
| FASTAPI | 137361 | 307 | 1983 | 0.93 | 1.7 | WEB/API | 2018/12/08 |
| POETRY | 153087 | 1022 | 3556 | 1.00 | 1.32 | DEVELOPER TOOLS | 2018/02/28 |
| HAYSTACK | 209305 | 775 | 2035 | 0.87 | 1.57 | ML/AI | 2019/11/14 |
| RICH | 244573 | 948 | 518 | 0.95 | 1.21 | DEVELOPER TOOLS | 2019/11/10 |
| PYDANTIC | 359486 | 1780 | 565 | 0.98 | 1.26 | DEVELOPER TOOLS | 2017/05/03 |
| PHIDATA | 552011 | 1799 | 2687 | 0.93 | 1.23 | DEVELOPER TOOLS | 2022/05/04 |
| PILLOW | 607377 | 3460 | 3923 | 0.99 | 1.35 | DATA SCIENCE | 2012/07/24 |
| SPHINX | 612272 | 4647 | 10285 | 1.00 | 1.27 | DEVELOPER TOOLS | 2015/01/02 |
| FACESWAP | 732572 | 3352 | 441 | 0.75 | 1.46 | OTHERS | 2017/12/19 |
| STREAMLIT | 806972 | 3236 | 3991 | 0.83 | 1.34 | WEB/API | 2019/08/24 |
| GRADIO | 874060 | 1845 | 1400 | 0.62 | 1.88 | WEB/API | 2018/12/19 |
| PIP | 1176602 | 5369 | 4789 | 0.62 | 1.36 | DEVELOPER TOOLS | 2011/03/06 |
| BLACK | 1497055 | 631 | 11892 | 0.99 | 1.28 | DEVELOPER TOOLS | 2018/03/14 |
| | | | VALIDATION | | | | |
| TYPER | 38236 | 167 | 1588 | 0.96 | 1.16 | DEVELOPER TOOLS | 2019/12/24 |
| PRE-COMMIT | 50380 | 345 | 2391 | 1.00 | 1.42 | DEVELOPER TOOLS | 2014/03/13 |
| FLASK | 73148 | 364 | 834 | 1.00 | 1.51 | WEB/API | 2010/04/06 |
| PDM | 172537 | 1291 | 532 | 0.98 | 1.37 | DEVELOPER TOOLS | 2019/12/27 |
| MANIM | 179218 | 1733 | 415 | 0.82 | 1.29 | DATA SCIENCE | 2020/05/19 |
| NICEGUI | 206638 | 1217 | 6018 | 0.98 | 1.09 | WEB/API | 2021/05/07 |
| OPENAI-PYTHON | 274146 | 1085 | 263 | 1.00 | 1.35 | ML/AI | 2020/10/25 |
| TAIPY | 403846 | 2356 | 2064 | 0.75 | 1.41 | WEB/API | 2022/02/18 |
| OPENBB | 1290501 | 3771 | 3137 | 0.70 | 1.8 | DATA SCIENCE | 2020/12/20 |
| CAPA | 1349016 | 1659 | 2935 | 0.64 | 1.53 | SECURITY | 2020/06/16 |
| | | | TEST1 | | | | |
| PRIVATE-GPT | 45562 | 197 | 257 | 0.98 | 1.36 | ML/AI | 2023/05/02 |
| GPTME | 58715 | 319 | 512 | 0.79 | 1.51 | ML/AI | 2023/03/24 |
| PAPER-QA | 73284 | 353 | 764 | 0.95 | 1.64 | ML/AI | 2023/02/05 |
| PANDAS-AI | 127754 | 996 | 1145 | 0.66 | 1.21 | ML/AI | 2023/04/22 |
| SUPERVISION | 150793 | 505 | 1101 | 0.90 | 1.49 | DATA SCIENCE | 2022/11/28 |
| GPT4FREE | 168395 | 679 | 808 | 0.76 | 1.11 | ML/AI | 2023/03/29 |
| AUTOGPT | 306046 | 1797 | 2235 | 0.78 | 1.34 | ML/AI | 2023/03/16 |
| MLC-LLM | 384359 | 1698 | 3182 | 0.83 | 1.25 | ML/AI | 2023/04/29 |
| DB-GPT | 817402 | 5329 | 9732 | 0.82 | 1.26 | ML/AI | 2023/04/13 |
| VLLM | 1037766 | 5271 | 12064 | 0.88 | 1.22 | ML/AI | 2023/02/09 |
| | | | TEST2 | | | | |
| SCREENSHOT-TO-CODE | 44482 | 60 | 102 | 0.73 | 1.58 | OTHERS | 2023/11/14 |
| EXO | 69991 | 406 | 721 | 0.61 | 1.28 | OTHERS | 2024/06/24 |
| TEN-AGENT | 71448 | 412 | 1076 | 0.76 | 1.07 | OTHERS | 2024/06/19 |
| GPT-PILOT | 94918 | 516 | 1101 | 0.82 | 1.19 | ML/AI | 2023/08/16 |
| APPWORLD | 156441 | 1125 | 2185 | 0.90 | 1.42 | OTHERS | 2024/06/23 |
| AGENTS | 156679 | 966 | 1956 | 0.88 | 1.14 | OTHERS | 2023/10/19 |
| LEROBOT | 183740 | 612 | 1068 | 0.57 | 1.69 | OTHERS | 2024/01/26 |
| LLAMA-FACTORY | 194043 | 520 | 1778 | 0.88 | 1.52 | ML/AI | 2023/05/28 |
| COMPOSIO | 345846 | 1059 | 1963 | 0.90 | 1.44 | DATA SCIENCE | 2024/02/23 |
| UNSTRACT | 495361 | 2281 | 2416 | 0.89 | 1.13 | DATA SCIENCE | 2024/02/21 |

*Table 6.* Repositories and their TypeCheck Errors. The repositories with slow `mypy` speed are reported with N/A.

| Train | | Dev | | Test1 | | Test2 | |
|---|---|---|---|---|---|---|---|
| Repo | Errors | Repo | Errors | Repo | Errors | Repo | Errors |
| pre-commit-hooks | 0 | pre-commit | 0 | gptme | 0 | screenshot-to-code | 0 |
| flake8 | 0 | flask | 0 | private-gpt | 14 | composio | 0 |
| urllib3 | 0 | nicegui | 2 | paper-qa | 24 | agents | 6 |
| haystack | 0 | typer | 4 | AutoGPT | 48 | unstract | 62 |
| rich | 0 | OpenBB | 4 | mlc-llm | 63 | TEN-Agent | 87 |
| black | 0 | openai-python | 8 | supervision | 136 | exo | 89 |
| fastapi | 1 | taipy | 52 | pandas-ai | 452 | LLaMA-Factory | 182 |
| Pillow | 1 | pdm | 76 | vllm | 466 | gpt-pilot | 332 |
| poetry | 2 | capa | 151 | gpt4free | 795 | lerobot | N/A |
| spotify-downloader | 9 | manim | N/A | DB-GPT | 1222 | appworld | N/A |
| faceswap | 11 | | | | | | |
| streamlit | 13 | | | | | | |
| deepface | 24 | | | | | | |
| phidata | 34 | | | | | | |
| vnpy | 137 | | | | | | |
| pip | 165 | | | | | | |
| sphinx | 218 | | | | | | |
| pydantic | 339 | | | | | | |
| GHunt | 589 | | | | | | |
| gradio | 846 | | | | | | |
| Average | 119.45 | Average | 33.0 | Average | 322.0 | Average | 94.75 |

*Table 7.* A list of similar type pairs. It can be seen that the TYPESIM score reasonably exhibits the similarity between the two types.

| Original | Predicted | TypeSim |
|---|---|---|
| `list[Any] | None` | `list[str] | None` | 0.75 |
| `dict[Any, Any] | None` | `dict[str, Any] | None` | 0.875 |
| `dict[str, tuple[Any, ...]]  | None` | `dict[str, tuple[int, ...]]  | None` | 0.9375 |
| `Any | dict[str, dict[Any, Any]]` | `Any | dict[str, Any]` | 0.875 |
| `dict[str, list[int]]` | `dict[str, Union[tuple[int, ...], Any]]` | 0.8385 |
| `float | np.ndarray[Any, Any]` | `np.ndarray[Any, Any]` | 0.5 |
| `pathlib.Path | None` | `str | pathlib.Path | None` | 0.6667 |

Figure 10 illustrates the prompt used for full-repo context type inference in our experiment. We first input the structure of the repository and followed by the content of the source code. Additionally, we specifies the expected answer and path format.

### C.2. TYPECHECK Score Calculation

We classify the following Mypy error types as indicators of consistency score: `attr-defined`, `assignment`, `arg-type`, `union-attr`, and `index`.

## D. Supplementary Experimental Results

### D.1. Results on Test Set

We further summarize the results for the test set in Table 8.

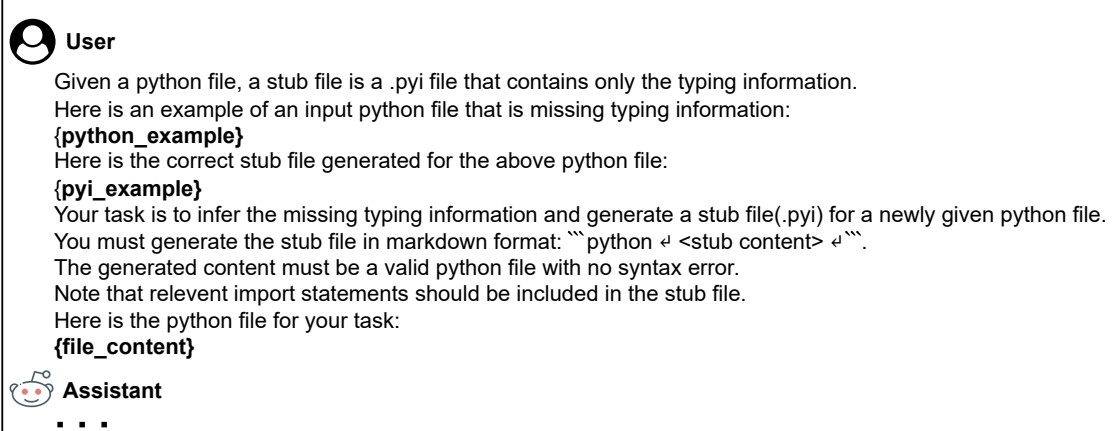

*Figure 9.* Prompt Template for Single File Context

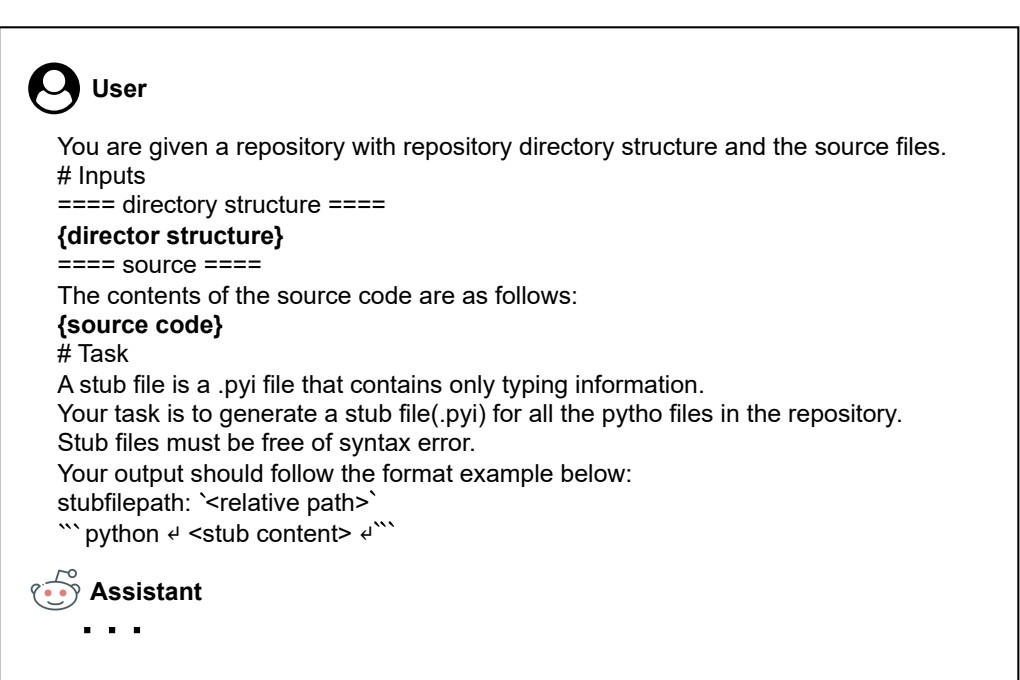

*Figure 10.* Prompt Template for Whole Repo Context

## D.2. Results for Each Model

We present the detailed results in Table 9 - 17 for each model on all repositories.

*Table 8.* Average TYPESIM scores and TYPECHECK scores of the repositories in the test set for various models.

| MODEL | TYPECHECK ↓ | TYPESIM ↑ | TYPESIM WO MISING ↑ | MISSING RATE ↓ |
|---|---|---|---|---|
| LLAMA-3-8B | 150.6 | 0.396 | 0.747 | 0.470 |
| LLAMA-3.1-8B | 400.2 | 0.634 | 0.815 | 0.225 |
| QWEN-2.5-7B | 405.7 | 0.649 | 0.817 | 0.207 |
| GPT-4O | 367.9 | 0.787 | 0.883 | 0.108 |
| GPT-4O-MINI | 251.7 | 0.779 | 0.882 | 0.117 |
| CLAUDE-3.5-SONNET | 173.3 | 0.790 | 0.876 | 0.098 |
| DEEPSEEK-V2.5 | 301.4 | 0.748 | 0.893 | 0.161 |
| DEEPSEEK-V3 | 192 | 0.793 | 0.889 | 0.107 |
| GROK-2 | 199.5 | 0.769 | 0.885 | 0.131 |

*Table 9.* TYPECHECK and TYPESIM scores for GPT-4O on each repository.

| REPO | TYPE CHECK | TYPE SIM | TYPESIM WO MISSING | MISSING RATIO | TYPESIM BY DEPTH | | | | |
|---|---|---|---|---|---|---|---|---|---|
| | | | | | DEPTH1 | DEPTH2 | DEPTH3 | DEPTH4 | DEPTH5 |
| AGENTS | 202 | 0.774 | 0.883 | 0.123 | 0.766 | 0.814 | 0.694 | 0.000 | |
| APPWORLD | N/A | 0.835 | 0.897 | 0.068 | 0.876 | 0.747 | 0.729 | 0.742 | 1.000 |
| AUTOGPT | 235 | 0.755 | 0.801 | 0.057 | 0.746 | 0.786 | 0.728 | 0.794 | 0.809 |
| BLACK | 131 | 0.807 | 0.898 | 0.101 | 0.822 | 0.771 | 0.692 | 0.888 | |
| CAPA | 546 | 0.883 | 0.902 | 0.021 | 0.922 | 0.786 | 0.781 | 0.817 | |
| COMPOSIO | 107 | 0.646 | 0.665 | 0.029 | 0.602 | 0.722 | 0.819 | 0.908 | |
| DB-GPT | 2482 | 0.803 | 0.874 | 0.081 | 0.808 | 0.792 | 0.822 | 0.790 | 0.781 |
| DEEPFACE | 4 | 0.898 | 0.963 | 0.068 | 0.886 | 0.940 | 0.875 | 0.990 | 1.000 |
| EXO | 66 | 0.746 | 0.928 | 0.196 | 0.729 | 0.810 | 0.666 | 0.875 | 0.750 |
| FACESWAP | 67 | 0.829 | 0.973 | 0.149 | 0.863 | 0.745 | 0.595 | 0.600 | 0.737 |
| FASTAPI | 290 | 0.810 | 0.876 | 0.075 | 0.893 | 0.746 | 0.782 | 0.917 | 0.930 |
| FLAKE8 | 16 | 0.793 | 0.936 | 0.153 | 0.824 | 0.720 | 0.765 | 1.000 | |
| FLASK | 70 | 0.876 | 0.931 | 0.059 | 0.903 | 0.863 | 0.576 | 0.424 | |
| GHUNT | 1 | 0.755 | 0.860 | 0.122 | 0.734 | 0.732 | 0.975 | 0.997 | |
| GPT-PILOT | 267 | 0.813 | 0.885 | 0.082 | 0.828 | 0.812 | 0.692 | 0.851 | |
| GPT4FREE | 413 | 0.835 | 0.898 | 0.071 | 0.873 | 0.691 | 0.695 | 0.965 | |
| GPTME | 84 | 0.877 | 0.926 | 0.053 | 0.875 | 0.869 | 0.963 | | 0.984 |
| GRADIO | 1161 | 0.655 | 0.836 | 0.216 | 0.715 | 0.629 | 0.546 | 0.457 | 0.709 |
| HAYSTACK | 335 | 0.692 | 0.910 | 0.239 | 0.675 | 0.696 | 0.755 | 0.871 | |
| LEROBOT | N/A | 0.785 | 0.903 | 0.131 | 0.778 | 0.806 | 0.761 | 0.730 | 0.892 |
| LLAMA-FACTORY | 168 | 0.820 | 0.903 | 0.092 | 0.858 | 0.740 | 0.789 | 0.783 | 1.000 |
| MANIM | N/A | 0.835 | 0.943 | 0.114 | 0.887 | 0.691 | 0.562 | 0.573 | 0.000 |
| MLC-LLM | 59 | 0.739 | 0.919 | 0.196 | 0.692 | 0.874 | 0.858 | 0.886 | 0.742 |
| NICEGUI | 426 | 0.828 | 0.863 | 0.040 | 0.844 | 0.786 | 0.726 | 0.871 | 0.403 |
| OPENAI-PYTHON | 434 | 0.696 | 0.783 | 0.110 | 0.744 | 0.633 | 0.704 | 0.167 | |
| OPENBB | 45 | 0.840 | 0.857 | 0.020 | 0.817 | 0.853 | 0.791 | 0.775 | 0.963 |
| PANDAS-AI | 209 | 0.729 | 0.787 | 0.073 | 0.737 | 0.731 | 0.663 | 0.776 | 1.000 |
| PAPER-QA | 143 | 0.867 | 0.904 | 0.041 | 0.911 | 0.846 | 0.726 | 0.825 | 0.943 |
| PDM | 380 | 0.880 | 0.932 | 0.056 | 0.920 | 0.773 | 0.754 | 0.839 | |
| PHIDATA | 1333 | 0.883 | 0.895 | 0.014 | 0.908 | 0.844 | 0.871 | 0.770 | 0.818 |
| PILLOW | 217 | 0.715 | 0.877 | 0.185 | 0.754 | 0.611 | 0.514 | 0.530 | 0.323 |
| PIP | 976 | 0.816 | 0.936 | 0.128 | 0.851 | 0.720 | 0.686 | 0.709 | 0.857 |
| POETRY | 93 | 0.864 | 0.957 | 0.097 | 0.909 | 0.742 | 0.663 | 0.431 | 0.958 |
| PRE-COMMIT | 33 | 0.900 | 0.910 | 0.011 | 0.910 | 0.872 | 0.951 | | |
| PRE-COMMIT-HOOKS | 1 | 0.942 | 0.942 | 0.000 | 0.934 | 0.945 | 0.973 | | |
| PRIVATE-GPT | 17 | 0.855 | 0.924 | 0.075 | 0.846 | 0.886 | 0.801 | | |
| PYDANTIC | 1143 | 0.815 | 0.887 | 0.081 | 0.875 | 0.700 | 0.647 | 0.701 | 0.000 |
| RICH | 45 | 0.785 | 0.957 | 0.179 | 0.794 | 0.756 | 0.688 | 0.969 | |
| SCREENSHOT-TO-CODE | 6 | 0.915 | 0.915 | 0.000 | 1.000 | 0.847 | 0.611 | 1.000 | |
| SPHINX | 1668 | 0.643 | 0.836 | 0.231 | 0.631 | 0.708 | 0.718 | 0.697 | 0.588 |
| SPOTIFY-DOWNLOADER | 59 | 0.876 | 0.911 | 0.039 | 0.887 | 0.837 | 0.947 | 0.913 | |
| STREAMLIT | 29 | 0.804 | 0.882 | 0.089 | 0.807 | 0.708 | 0.941 | | |
| SUPERVISION | 47 | 0.872 | 0.971 | 0.102 | 0.865 | 0.876 | 0.900 | 0.851 | 0.917 |
| TAIPY | 804 | 0.742 | 0.883 | 0.159 | 0.779 | 0.671 | 0.706 | 0.787 | 0.875 |
| TEN-AGENT | 113 | 0.698 | 0.833 | 0.163 | 0.683 | 0.776 | 0.720 | 0.802 | |
| TYPER | 94 | 0.701 | 0.937 | 0.252 | 0.761 | 0.625 | 0.617 | 0.000 | |
| UNSTRACT | 251 | 0.812 | 0.867 | 0.064 | 0.827 | 0.788 | 0.779 | 0.475 | |
| URLLIB3 | 117 | 0.819 | 0.901 | 0.091 | 0.877 | 0.723 | 0.749 | 0.962 | 1.000 |
| VLLM | 1770 | 0.759 | 0.885 | 0.143 | 0.764 | 0.758 | 0.704 | 0.681 | 0.972 |
| VNPY | 46 | 0.907 | 0.909 | 0.002 | 0.904 | 0.933 | 0.976 | 0.583 | |

*Table 10.* TYPECHECK and TYPESIM scores for GPT-4O-MINI on each repository.

| REPO | TYPE CHECK | TYPE SIM | TYPESIM WO MISSING | MISSING RATIO | TYPESIM BY DEPTH | | | | |
|------|------------|----------|--------------------|---------------|------------------|--------|--------|--------|--------|
| | | | | | DEPTH1 | DEPTH2 | DEPTH3 | DEPTH4 | DEPTH5 |
| AGENTS | 53 | 0.756 | 0.887 | 0.148 | 0.751 | 0.793 | 0.599 | 0.000 | |
| APPWORLD | N/A | 0.571 | 0.921 | 0.380 | 0.626 | 0.429 | 0.512 | 0.496 | 1.000 |
| AUTOGPT | 77 | 0.786 | 0.848 | 0.073 | 0.802 | 0.763 | 0.687 | 0.679 | 0.807 |
| BLACK | 157 | 0.731 | 0.881 | 0.171 | 0.759 | 0.635 | 0.615 | 0.913 | |
| CAPA | 332 | 0.762 | 0.806 | 0.055 | 0.793 | 0.732 | 0.644 | 0.817 | |
| COMPOSIO | 98 | 0.732 | 0.779 | 0.061 | 0.738 | 0.724 | 0.663 | 0.983 | |
| DB-GPT | 2033 | 0.815 | 0.886 | 0.080 | 0.819 | 0.809 | 0.815 | 0.803 | 0.818 |
| DEEPFACE | 9 | 0.894 | 0.949 | 0.058 | 0.898 | 0.898 | 0.876 | 0.990 | 0.688 |
| EXO | 74 | 0.730 | 0.901 | 0.190 | 0.713 | 0.783 | 0.699 | 0.859 | 0.719 |
| FACESWAP | 91 | 0.835 | 0.978 | 0.146 | 0.867 | 0.755 | 0.632 | 0.583 | 0.571 |
| FASTAPI | 722 | 0.777 | 0.861 | 0.098 | 0.872 | 0.725 | 0.688 | 0.824 | 0.922 |
| FLAKE8 | 28 | 0.776 | 0.923 | 0.159 | 0.837 | 0.656 | 0.501 | 1.000 | |
| FLASK | 19 | 0.784 | 0.922 | 0.150 | 0.833 | 0.704 | 0.565 | 0.236 | |
| GHUNT | 0 | 0.750 | 0.854 | 0.122 | 0.735 | 0.717 | 0.966 | 0.944 | |
| GPT-PILOT | 66 | 0.866 | 0.907 | 0.045 | 0.895 | 0.849 | 0.712 | 0.942 | |
| GPT4FREE | 341 | 0.796 | 0.886 | 0.102 | 0.836 | 0.643 | 0.655 | 0.975 | |
| GPTME | 28 | 0.888 | 0.917 | 0.031 | 0.897 | 0.860 | 0.888 | | 0.984 |
| GRADIO | 1088 | 0.629 | 0.834 | 0.246 | 0.684 | 0.612 | 0.510 | 0.397 | 0.580 |
| HAYSTACK | 448 | 0.707 | 0.915 | 0.227 | 0.694 | 0.699 | 0.799 | 0.660 | |
| LEROBOT | N/A | 0.752 | 0.896 | 0.160 | 0.767 | 0.777 | 0.623 | 0.577 | 0.892 |
| LLAMA-FACTORY | 119 | 0.802 | 0.865 | 0.074 | 0.834 | 0.751 | 0.709 | 0.738 | 0.994 |
| MANIM | N/A | 0.799 | 0.926 | 0.137 | 0.849 | 0.670 | 0.450 | 0.502 | 0.000 |
| MLC-LLM | 124 | 0.730 | 0.919 | 0.207 | 0.681 | 0.878 | 0.817 | 0.748 | 0.756 |
| NICEGUI | 318 | 0.831 | 0.898 | 0.075 | 0.859 | 0.741 | 0.683 | 0.862 | 0.482 |
| OPENAI-PYTHON | 633 | 0.726 | 0.807 | 0.100 | 0.704 | 0.760 | 0.691 | 0.500 | |
| OPENBB | 277 | 0.898 | 0.914 | 0.018 | 0.881 | 0.914 | 0.774 | 0.809 | 0.946 |
| PANDAS-AI | 170 | 0.731 | 0.804 | 0.091 | 0.732 | 0.760 | 0.623 | 0.786 | 1.000 |
| PAPER-QA | 21 | 0.811 | 0.881 | 0.080 | 0.850 | 0.769 | 0.741 | 0.863 | 0.948 |
| PDM | 230 | 0.859 | 0.909 | 0.055 | 0.917 | 0.711 | 0.640 | 0.642 | |
| PHIDATA | 1552 | 0.899 | 0.916 | 0.018 | 0.950 | 0.829 | 0.822 | 0.729 | 0.768 |
| PILLOW | 218 | 0.720 | 0.918 | 0.216 | 0.772 | 0.561 | 0.527 | 0.492 | 0.461 |
| PIP | 718 | 0.768 | 0.928 | 0.173 | 0.801 | 0.673 | 0.624 | 0.717 | 0.835 |
| POETRY | 67 | 0.855 | 0.951 | 0.101 | 0.907 | 0.712 | 0.561 | 0.513 | 0.825 |
| PRE-COMMIT | 19 | 0.879 | 0.900 | 0.024 | 0.913 | 0.796 | 0.881 | | |
| PRE-COMMIT-HOOKS | 12 | 0.905 | 0.905 | 0.000 | 0.920 | 0.930 | 0.808 | | |
| PRIVATE-GPT | 79 | 0.819 | 0.905 | 0.095 | 0.829 | 0.824 | 0.690 | | |
| PYDANTIC | 765 | 0.699 | 0.879 | 0.204 | 0.749 | 0.594 | 0.620 | 0.472 | 0.000 |
| RICH | 86 | 0.742 | 0.927 | 0.200 | 0.760 | 0.681 | 0.631 | 0.719 | |
| SCREENSHOT-TO-CODE | 3 | 0.905 | 0.905 | 0.000 | 0.969 | 0.855 | 0.616 | 1.000 | |
| SPHINX | 1157 | 0.624 | 0.808 | 0.227 | 0.622 | 0.645 | 0.627 | 0.475 | 0.525 |
| SPOTIFY-DOWNLOADER | 52 | 0.837 | 0.902 | 0.073 | 0.820 | 0.860 | 0.919 | 0.938 | |
| STREAMLIT | 21 | 0.877 | 0.932 | 0.060 | 0.882 | 0.763 | 0.973 | | |
| SUPERVISION | 56 | 0.796 | 0.966 | 0.175 | 0.848 | 0.727 | 0.726 | 0.854 | 0.301 |
| TAIPY | 704 | 0.760 | 0.878 | 0.134 | 0.809 | 0.667 | 0.712 | 0.784 | 0.813 |
| TEN-AGENT | 28 | 0.716 | 0.838 | 0.145 | 0.713 | 0.743 | 0.699 | 0.583 | |
| TYPER | 132 | 0.902 | 0.932 | 0.032 | 0.955 | 0.830 | 0.840 | 0.958 | |
| UNSTRACT | 91 | 0.807 | 0.864 | 0.066 | 0.817 | 0.784 | 0.811 | 0.913 | |
| URLLIB3 | 70 | 0.820 | 0.909 | 0.097 | 0.876 | 0.737 | 0.727 | 0.655 | 1.000 |
| VLLM | 1136 | 0.677 | 0.874 | 0.226 | 0.671 | 0.710 | 0.605 | 0.521 | 0.394 |
| VNPY | 75 | 0.934 | 0.934 | 0.000 | 0.931 | 0.933 | 0.973 | 1.000 | |

*Table 11.* TYPECHECK and TYPESIM scores for CLAUDE-3.5-SONNET on each repository.

| REPO | TYPE CHECK | TYPE SIM | TYPESIM WO MISSING | MISSING RATIO | TYPESIM BY DEPTH | | | | |
|---|---|---|---|---|---|---|---|---|---|
| | | | | | DEPTH1 | DEPTH2 | DEPTH3 | DEPTH4 | DEPTH5 |
| AGENTS | 21 | 0.646 | 0.852 | 0.242 | 0.643 | 0.673 | 0.514 | 0.000 | |
| APPWORLD | N/A | 0.660 | 0.927 | 0.287 | 0.725 | 0.502 | 0.584 | 0.512 | 0.000 |
| AUTOGPT | 60 | 0.769 | 0.856 | 0.102 | 0.749 | 0.821 | 0.793 | 0.752 | 0.604 |
| BLACK | 14 | 0.767 | 0.925 | 0.171 | 0.785 | 0.741 | 0.531 | 0.888 | |
| CAPA | 61 | 0.887 | 0.915 | 0.031 | 0.898 | 0.836 | 0.879 | 0.821 | |
| COMPOSIO | 12 | 0.578 | 0.684 | 0.155 | 0.569 | 0.581 | 0.634 | 0.960 | |
| DB-GPT | 1767 | 0.797 | 0.870 | 0.084 | 0.809 | 0.777 | 0.809 | 0.802 | 0.806 |
| DEEPFACE | 10 | 0.930 | 0.955 | 0.027 | 0.934 | 0.941 | 0.886 | 0.990 | 1.000 |
| EXO | 36 | 0.825 | 0.897 | 0.080 | 0.810 | 0.869 | 0.799 | 0.984 | 0.930 |
| FACESWAP | 17 | 0.824 | 0.965 | 0.146 | 0.848 | 0.770 | 0.651 | 0.650 | 0.571 |
| FASTAPI | 137 | 0.780 | 0.887 | 0.120 | 0.851 | 0.732 | 0.770 | 0.754 | 1.000 |
| FLAKE8 | 5 | 0.830 | 0.903 | 0.080 | 0.858 | 0.774 | 0.709 | 1.000 | |
| FLASK | 17 | 0.889 | 0.948 | 0.062 | 0.909 | 0.858 | 0.758 | 0.757 | |
| GHUNT | 11 | 0.853 | 0.853 | 0.000 | 0.920 | 0.703 | 0.975 | 0.993 | |
| GPT-PILOT | 93 | 0.848 | 0.886 | 0.043 | 0.856 | 0.848 | 0.765 | 0.973 | |
| GPT4FREE | 267 | 0.835 | 0.887 | 0.058 | 0.877 | 0.690 | 0.641 | 0.960 | |
| GPTME | 17 | 0.906 | 0.915 | 0.010 | 0.907 | 0.891 | 0.969 | | 0.984 |
| GRADIO | 150 | 0.722 | 0.846 | 0.147 | 0.774 | 0.701 | 0.629 | 0.518 | 0.700 |
| HAYSTACK | 462 | 0.726 | 0.889 | 0.183 | 0.708 | 0.705 | 0.876 | 0.788 | |
| LEROBOT | N/A | 0.822 | 0.916 | 0.103 | 0.819 | 0.833 | 0.788 | 0.867 | 0.892 |
| LLAMA-FACTORY | 73 | 0.746 | 0.874 | 0.147 | 0.771 | 0.697 | 0.697 | 0.797 | 1.000 |
| MANIM | N/A | 0.752 | 0.934 | 0.195 | 0.779 | 0.687 | 0.560 | 0.423 | 0.750 |
| MLC-LLM | 8 | 0.699 | 0.901 | 0.224 | 0.688 | 0.759 | 0.576 | 0.569 | 0.625 |
| NICEGUI | 123 | 0.806 | 0.870 | 0.074 | 0.822 | 0.782 | 0.567 | 0.843 | 0.369 |
| OPENAI-PYTHON | 187 | 0.519 | 0.804 | 0.355 | 0.515 | 0.518 | 0.603 | 0.799 | |
| OPENBB | 8 | 0.866 | 0.875 | 0.010 | 0.871 | 0.871 | 0.770 | 0.787 | 0.967 |
| PANDAS-AI | 105 | 0.759 | 0.773 | 0.018 | 0.800 | 0.708 | 0.648 | 0.804 | 0.969 |
| PAPER-QA | 7 | 0.830 | 0.897 | 0.075 | 0.860 | 0.804 | 0.758 | 0.907 | 0.943 |
| PDM | 89 | 0.763 | 0.870 | 0.122 | 0.789 | 0.692 | 0.701 | 0.777 | |
| PHIDATA | 339 | 0.928 | 0.939 | 0.012 | 0.961 | 0.887 | 0.828 | 0.922 | 0.827 |
| PILLOW | 20 | 0.754 | 0.916 | 0.178 | 0.776 | 0.682 | 0.680 | 0.597 | 0.915 |
| PIP | 264 | 0.809 | 0.921 | 0.122 | 0.838 | 0.725 | 0.661 | 0.820 | 0.857 |
| POETRY | 31 | 0.863 | 0.960 | 0.102 | 0.892 | 0.779 | 0.722 | 0.734 | 0.825 |
| PRE-COMMIT | 4 | 0.884 | 0.915 | 0.035 | 0.890 | 0.869 | 0.883 | | |
| PRE-COMMIT-HOOKS | 1 | 0.914 | 0.914 | 0.000 | 0.884 | 0.964 | 0.981 | | |
| PRIVATE-GPT | 7 | 0.879 | 0.946 | 0.070 | 0.857 | 0.934 | 0.866 | | |
| PYDANTIC | 433 | 0.684 | 0.868 | 0.212 | 0.680 | 0.702 | 0.675 | 0.324 | 0.000 |
| RICH | 10 | 0.760 | 0.952 | 0.202 | 0.760 | 0.767 | 0.669 | 0.719 | |
| SCREENSHOT-TO-CODE | 0 | 0.937 | 0.937 | 0.000 | 0.970 | 0.924 | 0.611 | 1.000 | |
| SPHINX | 178 | 0.626 | 0.872 | 0.282 | 0.611 | 0.704 | 0.725 | 0.652 | 0.582 |
| SPOTIFY-DOWNLOADER | 16 | 0.845 | 0.896 | 0.057 | 0.841 | 0.836 | 0.957 | 0.938 | |
| STREAMLIT | 5 | 0.564 | 0.910 | 0.380 | 0.530 | 0.796 | 0.993 | | |
| SUPERVISION | 26 | 0.884 | 0.964 | 0.084 | 0.895 | 0.861 | 0.896 | 0.926 | 0.660 |
| TAIPY | 160 | 0.652 | 0.887 | 0.265 | 0.694 | 0.574 | 0.589 | 0.767 | 0.563 |
| TEN-AGENT | 0 | 0.799 | 0.841 | 0.049 | 0.801 | 0.810 | 0.728 | 0.531 | |
| TYPER | 40 | 0.674 | 0.862 | 0.218 | 0.713 | 0.616 | 0.669 | 0.000 | |
| UNSTRACT | 11 | 0.803 | 0.833 | 0.036 | 0.814 | 0.788 | 0.763 | 0.525 | |
| URLLIB3 | 30 | 0.817 | 0.927 | 0.119 | 0.857 | 0.734 | 0.890 | 0.954 | 1.000 |
| VLLM | 615 | 0.743 | 0.891 | 0.166 | 0.733 | 0.774 | 0.733 | 0.637 | 0.797 |
| VNPY | 28 | 0.928 | 0.928 | 0.000 | 0.925 | 0.930 | 0.970 | 1.000 | |

*Table 12.* TYPECHECK and TYPESIM scores for DEEPSEEK-V3 on each repository.

| REPO | TYPE CHECK | TYPE SIM | TYPESIM WO MISSING | MISSING RATIO | TYPESIM BY DEPTH | | | | |
|---|---|---|---|---|---|---|---|---|---|
| | | | | | DEPTH1 | DEPTH2 | DEPTH3 | DEPTH4 | DEPTH5 |
| AGENTS | 33 | 0.728 | 0.885 | 0.178 | 0.719 | 0.778 | 0.534 | 0.000 | |
| APPWORLD | N/A | 0.721 | 0.910 | 0.208 | 0.798 | 0.503 | 0.755 | 0.572 | 0.500 |
| AUTOGPT | 61 | 0.788 | 0.825 | 0.046 | 0.781 | 0.800 | 0.827 | 0.764 | 0.656 |
| BLACK | 114 | 0.899 | 0.929 | 0.032 | 0.914 | 0.891 | 0.640 | 0.975 | |
| CAPA | 133 | 0.817 | 0.869 | 0.059 | 0.875 | 0.740 | 0.614 | 0.773 | |
| COMPOSIO | 95 | 0.717 | 0.766 | 0.064 | 0.718 | 0.721 | 0.664 | 0.903 | |
| DB-GPT | 1705 | 0.816 | 0.895 | 0.088 | 0.828 | 0.797 | 0.831 | 0.761 | 0.839 |
| DEEPFACE | 16 | 0.829 | 0.951 | 0.128 | 0.807 | 0.907 | 0.818 | 0.656 | 0.500 |
| EXO | 41 | 0.739 | 0.898 | 0.178 | 0.720 | 0.802 | 0.699 | 0.859 | 0.734 |
| FACESWAP | 62 | 0.857 | 0.982 | 0.127 | 0.881 | 0.797 | 0.694 | 0.650 | 0.737 |
| FASTAPI | 591 | 0.433 | 0.772 | 0.439 | 0.424 | 0.495 | 0.339 | 0.073 | 0.680 |
| FLAKE8 | 8 | 0.824 | 0.931 | 0.115 | 0.856 | 0.751 | 0.763 | 1.000 | |
| FLASK | 18 | 0.856 | 0.927 | 0.077 | 0.882 | 0.829 | 0.639 | 0.528 | |
| GHUNT | 14 | 0.743 | 0.846 | 0.122 | 0.742 | 0.678 | 0.981 | 1.000 | |
| GPT-PILOT | 76 | 0.865 | 0.908 | 0.048 | 0.879 | 0.855 | 0.787 | 0.963 | |
| GPT4FREE | 376 | 0.810 | 0.881 | 0.080 | 0.851 | 0.664 | 0.650 | 0.913 | |
| GPTME | 23 | 0.895 | 0.926 | 0.033 | 0.884 | 0.930 | 0.888 | | 0.984 |
| GRADIO | 1518 | 0.681 | 0.842 | 0.190 | 0.723 | 0.673 | 0.567 | 0.503 | 0.691 |
| HAYSTACK | 142 | 0.738 | 0.935 | 0.210 | 0.727 | 0.726 | 0.827 | 0.881 | |
| LEROBOT | N/A | 0.796 | 0.916 | 0.131 | 0.785 | 0.821 | 0.795 | 0.676 | 0.892 |
| LLAMA-FACTORY | 29 | 0.844 | 0.907 | 0.069 | 0.862 | 0.813 | 0.815 | 0.712 | 1.000 |
| MANIM | N/A | 0.767 | 0.944 | 0.187 | 0.815 | 0.636 | 0.491 | 0.563 | 0.000 |
| MLC-LLM | 105 | 0.749 | 0.930 | 0.195 | 0.700 | 0.896 | 0.811 | 0.840 | 0.693 |
| NICEGUI | 86 | 0.837 | 0.907 | 0.078 | 0.851 | 0.801 | 0.710 | 0.884 | 0.643 |
| OPENAI-PYTHON | 381 | 0.665 | 0.822 | 0.190 | 0.627 | 0.716 | 0.668 | 0.799 | |
| OPENBB | 63 | 0.883 | 0.929 | 0.050 | 0.912 | 0.876 | 0.789 | 0.913 | 0.973 |
| PANDAS-AI | 151 | 0.718 | 0.792 | 0.093 | 0.721 | 0.722 | 0.652 | 0.846 | 0.969 |
| PAPER-QA | 19 | 0.867 | 0.915 | 0.052 | 0.913 | 0.817 | 0.799 | 0.896 | 0.958 |
| PDM | 100 | 0.858 | 0.920 | 0.067 | 0.906 | 0.731 | 0.702 | 0.642 | |
| PHIDATA | 1372 | 0.931 | 0.948 | 0.018 | 0.976 | 0.866 | 0.860 | 0.844 | 0.853 |
| PILLOW | 53 | 0.828 | 0.924 | 0.104 | 0.865 | 0.709 | 0.721 | 0.672 | 0.756 |
| PIP | 482 | 0.803 | 0.936 | 0.142 | 0.836 | 0.736 | 0.686 | 0.254 | 0.839 |
| POETRY | 24 | 0.876 | 0.966 | 0.094 | 0.916 | 0.770 | 0.670 | 0.458 | 0.825 |
| PRE-COMMIT | 35 | 0.897 | 0.908 | 0.012 | 0.911 | 0.855 | 0.954 | | |
| PRE-COMMIT-HOOKS | 0 | 0.912 | 0.923 | 0.012 | 0.898 | 0.910 | 0.977 | | |
| PRIVATE-GPT | 15 | 0.825 | 0.910 | 0.093 | 0.826 | 0.830 | 0.801 | | |
| PYDANTIC | 253 | 0.701 | 0.861 | 0.187 | 0.711 | 0.667 | 0.739 | 0.757 | 0.000 |
| RICH | 63 | 0.754 | 0.939 | 0.197 | 0.754 | 0.762 | 0.662 | 0.719 | |
| SCREENSHOT-TO-CODE | 2 | 0.927 | 0.927 | 0.000 | 1.000 | 0.872 | 0.611 | 1.000 | |
| SPHINX | 374 | 0.621 | 0.780 | 0.203 | 0.611 | 0.668 | 0.724 | 0.761 | 0.527 |
| SPOTIFY-DOWNLOADER | 35 | 0.865 | 0.921 | 0.060 | 0.854 | 0.882 | 0.916 | 0.950 | |
| STREAMLIT | 12 | 0.882 | 0.934 | 0.056 | 0.886 | 0.769 | 0.993 | | |
| SUPERVISION | 137 | 0.826 | 0.967 | 0.146 | 0.866 | 0.762 | 0.822 | 0.807 | 0.301 |
| TAIPY | 492 | 0.775 | 0.879 | 0.119 | 0.799 | 0.722 | 0.784 | 0.815 | 0.625 |
| TEN-AGENT | 5 | 0.631 | 0.832 | 0.242 | 0.628 | 0.651 | 0.670 | 0.406 | |
| TYPER | 106 | 0.675 | 0.815 | 0.171 | 0.737 | 0.579 | 0.675 | 0.000 | |
| UNSTRACT | 207 | 0.803 | 0.861 | 0.067 | 0.810 | 0.782 | 0.843 | 0.875 | |
| URLLIB3 | 26 | 0.846 | 0.922 | 0.082 | 0.878 | 0.805 | 0.767 | 0.641 | 1.000 |
| VLLM | 391 | 0.719 | 0.908 | 0.209 | 0.698 | 0.767 | 0.735 | 0.702 | 0.947 |
| VNPY | 43 | 0.897 | 0.911 | 0.015 | 0.896 | 0.918 | 0.970 | 0.375 | |

*Table 13.* TYPECHECK and TYPESIM scores for DEEPSEEK-V2.5 on each repository.

| REPO | TYPE CHECK | TYPE SIM | TYPESIM WO MISSING | MISSING RATIO | TYPESIM BY DEPTH | | | | |
|------|-----------|----------|--------------------|---------------|-------|-------|-------|-------|-------|
| | | | | | DEPTH1 | DEPTH2 | DEPTH3 | DEPTH4 | DEPTH5 |
| AGENTS | 120 | 0.715 | 0.894 | 0.200 | 0.720 | 0.720 | 0.550 | 0.000 | |
| APPWORLD | N/A | 0.542 | 0.918 | 0.409 | 0.608 | 0.385 | 0.438 | 0.440 | 0.000 |
| AUTOGPT | 161 | 0.764 | 0.796 | 0.040 | 0.763 | 0.774 | 0.735 | 0.791 | 0.754 |
| BLACK | 53 | 0.749 | 0.937 | 0.201 | 0.788 | 0.612 | 0.614 | 0.879 | |
| CAPA | 382 | 0.823 | 0.886 | 0.071 | 0.841 | 0.750 | 0.798 | 0.894 | |
| COMPOSIO | 32 | 0.671 | 0.757 | 0.114 | 0.683 | 0.659 | 0.569 | 0.813 | |
| DB-GPT | 2153 | 0.776 | 0.899 | 0.137 | 0.778 | 0.774 | 0.768 | 0.730 | 0.837 |
| DEEPFACE | 15 | 0.932 | 0.965 | 0.034 | 0.925 | 0.972 | 0.906 | 0.741 | 1.000 |
| EXO | 90 | 0.811 | 0.935 | 0.133 | 0.816 | 0.814 | 0.649 | 0.833 | 0.710 |
| FACESWAP | 111 | 0.802 | 0.969 | 0.172 | 0.838 | 0.716 | 0.561 | 0.467 | 0.737 |
| FASTAPI | 462 | 0.533 | 0.961 | 0.445 | 0.498 | 0.650 | 0.336 | 0.073 | 0.680 |
| FLAKE8 | 20 | 0.832 | 0.950 | 0.125 | 0.870 | 0.748 | 0.749 | 1.000 | |
| FLASK | 28 | 0.799 | 0.940 | 0.150 | 0.843 | 0.735 | 0.545 | 0.257 | |
| GHUNT | 47 | 0.766 | 0.880 | 0.129 | 0.742 | 0.751 | 0.974 | 1.000 | |
| GPT-PILOT | 129 | 0.871 | 0.911 | 0.045 | 0.894 | 0.856 | 0.754 | 0.947 | |
| GPT4FREE | 615 | 0.759 | 0.898 | 0.155 | 0.787 | 0.654 | 0.649 | 0.903 | |
| GPTME | 54 | 0.898 | 0.924 | 0.027 | 0.898 | 0.900 | 0.883 | | 0.984 |
| GRADIO | 1728 | 0.574 | 0.855 | 0.328 | 0.603 | 0.574 | 0.448 | 0.481 | 0.772 |
| HAYSTACK | 477 | 0.708 | 0.949 | 0.254 | 0.700 | 0.708 | 0.736 | 0.875 | |
| LEROBOT | N/A | 0.682 | 0.925 | 0.262 | 0.646 | 0.729 | 0.757 | 0.493 | 0.892 |
| LLAMA-FACTORY | 30 | 0.803 | 0.927 | 0.133 | 0.828 | 0.748 | 0.811 | 0.644 | 0.987 |
| MANIM | N/A | 0.635 | 0.870 | 0.270 | 0.687 | 0.481 | 0.398 | 0.493 | 0.000 |
| MLC-LLM | 169 | 0.575 | 0.951 | 0.396 | 0.537 | 0.696 | 0.574 | 0.698 | 0.601 |
| NICEGUI | 685 | 0.801 | 0.901 | 0.111 | 0.797 | 0.822 | 0.767 | 0.851 | 0.623 |
| OPENAI-PYTHON | 748 | 0.571 | 0.819 | 0.302 | 0.568 | 0.573 | 0.606 | 0.625 | |
| OPENBB | 193 | 0.790 | 0.927 | 0.148 | 0.836 | 0.778 | 0.707 | 0.671 | 0.758 |
| PANDAS-AI | 182 | 0.746 | 0.817 | 0.087 | 0.737 | 0.789 | 0.658 | 0.803 | 1.000 |
| PAPER-QA | 32 | 0.641 | 0.885 | 0.275 | 0.653 | 0.631 | 0.634 | 0.506 | 0.943 |
| PDM | 156 | 0.765 | 0.865 | 0.116 | 0.814 | 0.638 | 0.592 | 0.637 | |
| PHIDATA | 1050 | 0.883 | 0.939 | 0.059 | 0.914 | 0.848 | 0.810 | 0.823 | 0.443 |
| PILLOW | 201 | 0.630 | 0.857 | 0.265 | 0.666 | 0.530 | 0.471 | 0.387 | 0.484 |
| PIP | 694 | 0.788 | 0.925 | 0.148 | 0.820 | 0.724 | 0.664 | 0.253 | 0.741 |
| POETRY | 53 | 0.876 | 0.970 | 0.097 | 0.914 | 0.773 | 0.682 | 0.463 | 0.958 |
| PRE-COMMIT | 40 | 0.943 | 0.953 | 0.011 | 0.965 | 0.887 | 0.960 | | |
| PRE-COMMIT-HOOKS | 7 | 0.952 | 0.960 | 0.008 | 0.983 | 0.907 | 0.871 | | |
| PRIVATE-GPT | 44 | 0.751 | 0.923 | 0.186 | 0.764 | 0.736 | 0.686 | | |
| PYDANTIC | 751 | 0.667 | 0.912 | 0.268 | 0.714 | 0.562 | 0.631 | 0.431 | 0.000 |
| RICH | 291 | 0.618 | 0.934 | 0.339 | 0.631 | 0.563 | 0.659 | 0.719 | |
| SCREENSHOT-TO-CODE | 6 | 0.887 | 0.887 | 0.000 | 0.980 | 0.813 | 0.611 | 0.833 | |
| SPHINX | 666 | 0.498 | 0.840 | 0.407 | 0.476 | 0.625 | 0.610 | 0.513 | 0.590 |
| SPOTIFY-DOWNLOADER | 32 | 0.861 | 0.910 | 0.053 | 0.856 | 0.858 | 0.973 | 0.900 | |
| STREAMLIT | 98 | 0.910 | 0.980 | 0.071 | 0.916 | 0.782 | 1.000 | | |
| SUPERVISION | 172 | 0.803 | 0.969 | 0.171 | 0.847 | 0.747 | 0.756 | 0.762 | 0.315 |
| TAIPY | 773 | 0.790 | 0.897 | 0.120 | 0.810 | 0.750 | 0.775 | 0.783 | 0.875 |
| TEN-AGENT | 18 | 0.590 | 0.869 | 0.322 | 0.584 | 0.621 | 0.631 | 0.469 | |
| TYPER | 112 | 0.676 | 0.903 | 0.252 | 0.736 | 0.581 | 0.678 | 0.000 | |
| UNSTRACT | 84 | 0.812 | 0.872 | 0.068 | 0.823 | 0.793 | 0.788 | 0.738 | |
| URLLIB3 | 50 | 0.848 | 0.933 | 0.091 | 0.896 | 0.776 | 0.790 | 0.641 | 1.000 |
| VLLM | 1372 | 0.661 | 0.874 | 0.244 | 0.660 | 0.669 | 0.646 | 0.573 | 0.576 |
| VNPY | 63 | 0.898 | 0.914 | 0.018 | 0.895 | 0.940 | 0.976 | 0.375 | |

*Table 14.* TYPECHECK and TYPESIM scores for GROK-2 on each repository.

| REPO | TYPE CHECK | TYPE SIM | TYPESIM WO MISSING | MISSING RATIO | TYPESIM BY DEPTH | | | | |
|---|---|---|---|---|---|---|---|---|---|
| | | | | | DEPTH1 | DEPTH2 | DEPTH3 | DEPTH4 | DEPTH5 |
| AGENTS | 37 | 0.700 | 0.900 | 0.222 | 0.695 | 0.742 | 0.493 | 0.000 | |
| APPWORLD | N/A | 0.535 | 0.931 | 0.426 | 0.596 | 0.395 | 0.413 | 0.491 | 0.000 |
| AUTOGPT | 81 | 0.755 | 0.803 | 0.060 | 0.756 | 0.752 | 0.762 | 0.773 | 0.750 |
| BLACK | 22 | 0.825 | 0.925 | 0.107 | 0.847 | 0.807 | 0.489 | 0.958 | |
| CAPA | 89 | 0.905 | 0.924 | 0.021 | 0.936 | 0.794 | 0.851 | 0.852 | |
| COMPOSIO | 35 | 0.482 | 0.593 | 0.188 | 0.454 | 0.563 | 0.445 | 0.897 | |
| DB-GPT | 2066 | 0.820 | 0.898 | 0.086 | 0.834 | 0.805 | 0.789 | 0.764 | 0.841 |
| DEEPFACE | 5 | 0.868 | 0.961 | 0.096 | 0.862 | 0.915 | 0.808 | 0.990 | 1.000 |
| EXO | 49 | 0.778 | 0.928 | 0.161 | 0.763 | 0.838 | 0.696 | 0.875 | 0.750 |
| FACESWAP | 64 | 0.885 | 0.973 | 0.091 | 0.904 | 0.833 | 0.784 | 0.817 | 0.737 |
| FASTAPI | 707 | 0.659 | 0.940 | 0.299 | 0.751 | 0.646 | 0.468 | 0.577 | 0.979 |
| FLAKE8 | 5 | 0.917 | 0.938 | 0.023 | 0.930 | 0.884 | 0.903 | 1.000 | |
| FLASK | 22 | 0.900 | 0.935 | 0.037 | 0.933 | 0.847 | 0.692 | 0.861 | |
| GHUNT | 12 | 0.751 | 0.894 | 0.160 | 0.732 | 0.735 | 0.921 | 1.000 | |
| GPT-PILOT | 20 | 0.839 | 0.891 | 0.059 | 0.843 | 0.841 | 0.777 | 0.948 | |
| GPT4FREE | 342 | 0.839 | 0.883 | 0.050 | 0.879 | 0.690 | 0.705 | 0.917 | |
| GPTME | 22 | 0.908 | 0.937 | 0.031 | 0.907 | 0.912 | 0.894 | | 0.984 |
| GRADIO | 328 | 0.695 | 0.861 | 0.192 | 0.739 | 0.677 | 0.612 | 0.569 | 0.630 |
| HAYSTACK | 514 | 0.747 | 0.933 | 0.199 | 0.712 | 0.773 | 0.837 | 0.867 | |
| LEROBOT | N/A | 0.851 | 0.922 | 0.077 | 0.866 | 0.860 | 0.764 | 0.704 | 0.892 |
| LLAMA-FACTORY | 48 | 0.837 | 0.889 | 0.059 | 0.864 | 0.785 | 0.797 | 0.775 | 0.987 |
| MANIM | N/A | 0.650 | 0.872 | 0.255 | 0.667 | 0.619 | 0.461 | 0.641 | 0.000 |
| MLC-LLM | 32 | 0.734 | 0.946 | 0.224 | 0.703 | 0.819 | 0.837 | 0.790 | 0.905 |
| NICEGUI | 36 | 0.772 | 0.871 | 0.114 | 0.765 | 0.815 | 0.688 | 0.853 | 0.576 |
| OPENAI-PYTHON | 208 | 0.693 | 0.814 | 0.149 | 0.690 | 0.691 | 0.776 | 0.713 | |
| OPENBB | 14 | 0.864 | 0.928 | 0.070 | 0.854 | 0.875 | 0.764 | 0.738 | 0.967 |
| PANDAS-AI | 86 | 0.741 | 0.817 | 0.093 | 0.745 | 0.759 | 0.642 | 0.833 | 1.000 |
| PAPER-QA | 15 | 0.857 | 0.903 | 0.051 | 0.904 | 0.808 | 0.791 | 0.828 | 0.838 |
| PDM | 86 | 0.870 | 0.920 | 0.055 | 0.919 | 0.741 | 0.701 | 0.688 | |
| PHIDATA | 1234 | 0.899 | 0.943 | 0.047 | 0.945 | 0.844 | 0.798 | 0.682 | 0.443 |
| PILLOW | 38 | 0.818 | 0.923 | 0.114 | 0.863 | 0.688 | 0.629 | 0.533 | 0.864 |
| PIP | 240 | 0.814 | 0.936 | 0.130 | 0.842 | 0.765 | 0.704 | 0.179 | 0.848 |
| POETRY | 38 | 0.867 | 0.961 | 0.098 | 0.909 | 0.752 | 0.652 | 0.519 | 0.825 |
| PRE-COMMIT | 2 | 0.886 | 0.946 | 0.063 | 0.895 | 0.858 | 0.966 | | |
| PRE-COMMIT-HOOKS | 2 | 0.905 | 0.905 | 0.000 | 0.888 | 0.924 | 0.952 | | |
| PRIVATE-GPT | 4 | 0.645 | 0.939 | 0.313 | 0.609 | 0.712 | 0.724 | | |
| PYDANTIC | 564 | 0.816 | 0.920 | 0.113 | 0.811 | 0.827 | 0.837 | 0.551 | 0.500 |
| RICH | 16 | 0.628 | 0.856 | 0.266 | 0.628 | 0.630 | 0.601 | 0.719 | |
| SCREENSHOT-TO-CODE | 1 | 0.936 | 0.936 | 0.000 | 0.990 | 0.904 | 0.616 | 0.833 | |
| SPHINX | 243 | 0.638 | 0.803 | 0.205 | 0.633 | 0.679 | 0.639 | 0.531 | 0.594 |
| SPOTIFY-DOWNLOADER | 21 | 0.868 | 0.920 | 0.057 | 0.864 | 0.867 | 0.969 | 0.788 | |
| STREAMLIT | 42 | 0.954 | 0.979 | 0.025 | 0.967 | 0.769 | 0.993 | | |
| SUPERVISION | 24 | 0.852 | 0.967 | 0.119 | 0.904 | 0.795 | 0.779 | 0.708 | 0.301 |
| TAIPY | 528 | 0.737 | 0.875 | 0.158 | 0.763 | 0.687 | 0.712 | 0.797 | 0.875 |
| TEN-AGENT | 2 | 0.468 | 0.912 | 0.487 | 0.475 | 0.425 | 0.561 | 0.323 | |
| TYPER | 175 | 0.690 | 0.923 | 0.252 | 0.744 | 0.608 | 0.689 | 0.000 | |
| UNSTRACT | 33 | 0.809 | 0.877 | 0.078 | 0.818 | 0.791 | 0.807 | 0.738 | |
| URLLIB3 | 30 | 0.854 | 0.910 | 0.062 | 0.902 | 0.774 | 0.862 | 0.608 | 1.000 |
| VLLM | 696 | 0.732 | 0.869 | 0.158 | 0.717 | 0.754 | 0.809 | 0.731 | 0.972 |
| VNPY | 56 | 0.860 | 0.913 | 0.057 | 0.871 | 0.600 | 0.976 | 1.000 | |

*Table 15.* TYPECHECK and TYPESIM scores for LLAMA-3-8B on each repository.

| REPO | TYPE CHECK | TYPE SIM | TYPESIM WO MISSING | MISSING RATIO | TYPESIM BY DEPTH | | | | |
|---|---|---|---|---|---|---|---|---|---|
| | | | | | DEPTH1 | DEPTH2 | DEPTH3 | DEPTH4 | DEPTH5 |
| AGENTS | 119 | 0.415 | 0.809 | 0.487 | 0.426 | 0.408 | 0.154 | 0.000 | |
| APPWORLD | N/A | 0.177 | 0.901 | 0.804 | 0.194 | 0.131 | 0.176 | 0.168 | 0.000 |
| AUTOGPT | 82 | 0.451 | 0.749 | 0.398 | 0.500 | 0.370 | 0.180 | 0.227 | 0.000 |
| BLACK | 30 | 0.186 | 0.805 | 0.769 | 0.201 | 0.133 | 0.142 | 0.388 | |
| CAPA | 161 | 0.235 | 0.606 | 0.612 | 0.246 | 0.147 | 0.258 | 0.219 | |
| COMPOSIO | 115 | 0.382 | 0.573 | 0.333 | 0.458 | 0.218 | 0.198 | 0.068 | |
| DB-GPT | 1032 | 0.317 | 0.604 | 0.476 | 0.381 | 0.228 | 0.266 | 0.289 | 0.216 |
| DEEPFACE | 14 | 0.231 | 0.630 | 0.634 | 0.298 | 0.141 | 0.107 | 0.000 | 0.000 |
| EXO | 52 | 0.377 | 0.782 | 0.519 | 0.390 | 0.343 | 0.341 | 0.208 | 0.458 |
| FACESWAP | 42 | 0.326 | 0.867 | 0.624 | 0.374 | 0.180 | 0.092 | 0.250 | 0.323 |
| FASTAPI | 1792 | 0.054 | 0.754 | 0.928 | 0.089 | 0.032 | 0.062 | 0.000 | 0.000 |
| FLAKE8 | 28 | 0.573 | 0.815 | 0.297 | 0.649 | 0.441 | 0.181 | 0.250 | |
| FLASK | 15 | 0.346 | 0.764 | 0.547 | 0.392 | 0.242 | 0.259 | 0.257 | |
| GHUNT | 21 | 0.481 | 0.849 | 0.434 | 0.543 | 0.338 | 0.690 | 0.005 | |
| GPT-PILOT | 68 | 0.574 | 0.768 | 0.253 | 0.620 | 0.523 | 0.438 | 0.828 | |
| GPT4FREE | 168 | 0.543 | 0.809 | 0.329 | 0.579 | 0.418 | 0.343 | 0.892 | |
| GPTME | 36 | 0.567 | 0.827 | 0.315 | 0.613 | 0.491 | 0.196 | | 0.000 |
| GRADIO | 772 | 0.227 | 0.576 | 0.605 | 0.326 | 0.169 | 0.084 | 0.018 | 0.150 |
| HAYSTACK | 166 | 0.308 | 0.558 | 0.448 | 0.463 | 0.114 | 0.135 | 0.188 | |
| LEROBOT | N/A | 0.216 | 0.754 | 0.714 | 0.309 | 0.096 | 0.147 | 0.079 | 0.000 |
| LLAMA-FACTORY | 60 | 0.366 | 0.730 | 0.498 | 0.401 | 0.325 | 0.225 | 0.106 | 0.828 |
| MANIM | N/A | 0.299 | 0.697 | 0.570 | 0.335 | 0.198 | 0.103 | 0.445 | 0.000 |
| MLC-LLM | 108 | 0.243 | 0.738 | 0.670 | 0.260 | 0.200 | 0.180 | 0.243 | 0.000 |
| NICEGUI | 272 | 0.514 | 0.811 | 0.367 | 0.527 | 0.506 | 0.269 | 0.424 | 0.100 |
| OPENAI-PYTHON | 296 | 0.317 | 0.668 | 0.525 | 0.358 | 0.270 | 0.213 | 0.354 | |
| OPENBB | 102 | 0.298 | 0.589 | 0.495 | 0.473 | 0.237 | 0.129 | 0.194 | 0.350 |
| PANDAS-AI | 87 | 0.471 | 0.585 | 0.196 | 0.558 | 0.396 | 0.205 | 0.250 | 0.000 |
| PAPER-QA | 37 | 0.286 | 0.715 | 0.601 | 0.352 | 0.201 | 0.255 | 0.197 | 0.000 |
| PDM | 268 | 0.413 | 0.781 | 0.472 | 0.477 | 0.246 | 0.187 | 0.116 | |
| PHIDATA | 636 | 0.443 | 0.629 | 0.296 | 0.593 | 0.236 | 0.173 | 0.117 | 0.000 |
| PILLOW | 26 | 0.382 | 0.885 | 0.569 | 0.427 | 0.254 | 0.181 | 0.142 | 0.053 |
| PIP | 253 | 0.440 | 0.861 | 0.489 | 0.478 | 0.356 | 0.245 | 0.058 | 0.134 |
| POETRY | 98 | 0.600 | 0.873 | 0.312 | 0.662 | 0.433 | 0.224 | 0.344 | 0.492 |
| PRE-COMMIT | 65 | 0.584 | 0.851 | 0.314 | 0.634 | 0.447 | 0.761 | | |
| PRE-COMMIT-HOOKS | 19 | 0.649 | 0.887 | 0.269 | 0.657 | 0.737 | 0.500 | | |
| PRIVATE-GPT | 13 | 0.410 | 0.814 | 0.497 | 0.386 | 0.440 | 0.513 | | |
| PYDANTIC | 223 | 0.113 | 0.652 | 0.826 | 0.140 | 0.051 | 0.091 | 0.111 | 0.000 |
| RICH | 49 | 0.170 | 0.664 | 0.744 | 0.191 | 0.091 | 0.173 | 0.000 | |
| SCREENSHOT-TO-CODE | 5 | 0.532 | 0.952 | 0.441 | 0.633 | 0.456 | 0.306 | 0.000 | |
| SPHINX | 229 | 0.097 | 0.620 | 0.844 | 0.097 | 0.096 | 0.083 | 0.103 | 0.207 |
| SPOTIFY-DOWNLOADER | 15 | 0.395 | 0.631 | 0.374 | 0.459 | 0.255 | 0.294 | 0.654 | |
| STREAMLIT | 237 | 0.079 | 0.232 | 0.658 | 0.077 | 0.111 | 0.059 | | |
| SUPERVISION | 48 | 0.203 | 0.644 | 0.685 | 0.260 | 0.114 | 0.189 | 0.142 | 0.000 |
| TAIPY | 144 | 0.357 | 0.589 | 0.394 | 0.501 | 0.129 | 0.088 | 0.029 | 0.188 |
| TEN-AGENT | 80 | 0.393 | 0.827 | 0.525 | 0.412 | 0.307 | 0.287 | 0.139 | |
| TYPER | 72 | 0.300 | 0.917 | 0.673 | 0.314 | 0.265 | 0.369 | 0.000 | |
| UNSTRACT | 182 | 0.573 | 0.759 | 0.245 | 0.638 | 0.435 | 0.518 | 0.763 | |
| URLLIB3 | 27 | 0.462 | 0.837 | 0.448 | 0.537 | 0.362 | 0.281 | 0.079 | 0.000 |
| VLLM | 413 | 0.185 | 0.531 | 0.651 | 0.214 | 0.132 | 0.112 | 0.072 | 0.169 |
| VNPY | 37 | 0.590 | 0.772 | 0.235 | 0.652 | 0.117 | 0.079 | 0.000 | |

*Table 16.* TYPECHECK and TYPESIM scores for LLAMA-3.1-8B on each repository.

| REPO | TYPE CHECK | TYPE SIM | TYPESIM WO MISSING | MISSING RATIO | TYPESIM BY DEPTH | | | | |
|---|---|---|---|---|---|---|---|---|---|
| | | | | | DEPTH1 | DEPTH2 | DEPTH3 | DEPTH4 | DEPTH5 |
| AGENTS | 153 | 0.636 | 0.809 | 0.214 | 0.634 | 0.669 | 0.410 | 0.000 | |
| APPWORLD | N/A | 0.347 | 0.789 | 0.560 | 0.381 | 0.258 | 0.326 | 0.392 | 0.000 |
| AUTOGPT | 273 | 0.676 | 0.783 | 0.137 | 0.698 | 0.629 | 0.604 | 0.679 | 0.500 |
| BLACK | 189 | 0.518 | 0.782 | 0.338 | 0.554 | 0.398 | 0.345 | 0.763 | |
| CAPA | 497 | 0.574 | 0.816 | 0.297 | 0.588 | 0.575 | 0.498 | 0.906 | |
| COMPOSIO | 283 | 0.580 | 0.693 | 0.163 | 0.626 | 0.495 | 0.377 | 0.847 | |
| DB-GPT | 2591 | 0.642 | 0.769 | 0.165 | 0.685 | 0.586 | 0.595 | 0.571 | 0.605 |
| DEEPFACE | 38 | 0.716 | 0.923 | 0.224 | 0.753 | 0.680 | 0.612 | 0.616 | 1.000 |
| EXO | 147 | 0.734 | 0.938 | 0.218 | 0.708 | 0.812 | 0.732 | 0.943 | 0.714 |
| FACESWAP | 199 | 0.710 | 0.908 | 0.219 | 0.745 | 0.629 | 0.427 | 0.533 | 0.722 |
| FASTAPI | 2260 | 0.275 | 0.573 | 0.519 | 0.231 | 0.359 | 0.175 | 0.008 | 0.000 |
| FLAKE8 | 40 | 0.757 | 0.882 | 0.142 | 0.813 | 0.653 | 0.488 | 0.750 | |
| FLASK | 85 | 0.639 | 0.772 | 0.173 | 0.687 | 0.556 | 0.363 | 0.444 | |
| GHUNT | 221 | 0.755 | 0.863 | 0.126 | 0.749 | 0.717 | 0.964 | 0.672 | |
| GPT-PILOT | 398 | 0.799 | 0.854 | 0.065 | 0.814 | 0.789 | 0.706 | 0.951 | |
| GPT4FREE | 608 | 0.747 | 0.885 | 0.156 | 0.782 | 0.621 | 0.588 | 0.957 | |
| GPTME | 106 | 0.586 | 0.850 | 0.311 | 0.607 | 0.531 | 0.504 | | 0.984 |
| GRADIO | 2414 | 0.456 | 0.593 | 0.231 | 0.552 | 0.404 | 0.299 | 0.249 | 0.296 |
| HAYSTACK | 450 | 0.243 | 0.721 | 0.663 | 0.274 | 0.212 | 0.183 | 0.288 | |
| LEROBOT | N/A | 0.537 | 0.830 | 0.352 | 0.694 | 0.337 | 0.446 | 0.150 | 0.250 |
| LLAMA-FACTORY | 280 | 0.628 | 0.794 | 0.209 | 0.650 | 0.597 | 0.574 | 0.389 | 0.961 |
| MANIM | N/A | 0.681 | 0.833 | 0.183 | 0.738 | 0.524 | 0.360 | 0.440 | 0.000 |
| MLC-LLM | 208 | 0.569 | 0.856 | 0.336 | 0.543 | 0.661 | 0.553 | 0.448 | 0.423 |
| NICEGUI | 213 | 0.579 | 0.824 | 0.297 | 0.573 | 0.612 | 0.557 | 0.652 | 0.507 |
| OPENAI-PYTHON | 906 | 0.539 | 0.702 | 0.233 | 0.572 | 0.504 | 0.405 | 0.608 | |
| OPENBB | 237 | 0.599 | 0.839 | 0.286 | 0.607 | 0.607 | 0.479 | 0.339 | 0.200 |
| PANDAS-AI | 431 | 0.670 | 0.779 | 0.141 | 0.673 | 0.723 | 0.507 | 0.601 | 0.875 |
| PAPER-QA | 135 | 0.595 | 0.742 | 0.198 | 0.675 | 0.486 | 0.592 | 0.376 | 0.948 |
| PDM | 422 | 0.601 | 0.800 | 0.249 | 0.671 | 0.427 | 0.306 | 0.160 | |
| PHIDATA | 1412 | 0.751 | 0.868 | 0.135 | 0.800 | 0.693 | 0.656 | 0.542 | 0.136 |
| PILLOW | 138 | 0.495 | 0.784 | 0.369 | 0.550 | 0.329 | 0.275 | 0.294 | 0.296 |
| PIP | 1161 | 0.594 | 0.831 | 0.285 | 0.631 | 0.505 | 0.475 | 0.201 | 0.272 |
| POETRY | 241 | 0.754 | 0.922 | 0.182 | 0.799 | 0.633 | 0.499 | 0.371 | 0.958 |
| PRE-COMMIT | 46 | 0.813 | 0.878 | 0.074 | 0.895 | 0.640 | 0.535 | | |
| PRE-COMMIT-HOOKS | 22 | 0.827 | 0.885 | 0.066 | 0.884 | 0.800 | 0.605 | | |
| PRIVATE-GPT | 69 | 0.750 | 0.809 | 0.073 | 0.852 | 0.559 | 0.542 | | |
| PYDANTIC | 786 | 0.290 | 0.565 | 0.486 | 0.358 | 0.142 | 0.198 | 0.111 | 0.000 |
| RICH | 309 | 0.458 | 0.705 | 0.350 | 0.464 | 0.433 | 0.521 | 0.250 | |
| SCREENSHOT-TO-CODE | 7 | 0.912 | 0.930 | 0.020 | 0.960 | 0.883 | 0.616 | 0.833 | |
| SPHINX | 319 | 0.022 | 0.537 | 0.959 | 0.021 | 0.030 | 0.016 | 0.029 | 0.000 |
| SPOTIFY-DOWNLOADER | 133 | 0.722 | 0.891 | 0.189 | 0.714 | 0.745 | 0.712 | 0.600 | |
| STREAMLIT | 327 | 0.593 | 0.885 | 0.330 | 0.569 | 0.722 | 0.980 | | |
| SUPERVISION | 306 | 0.508 | 0.875 | 0.420 | 0.465 | 0.562 | 0.559 | 0.653 | 0.000 |
| TAIPY | 834 | 0.571 | 0.769 | 0.256 | 0.623 | 0.495 | 0.405 | 0.663 | 0.438 |
| TEN-AGENT | 135 | 0.655 | 0.778 | 0.158 | 0.652 | 0.690 | 0.551 | 0.486 | |
| TYPER | 458 | 0.508 | 0.889 | 0.429 | 0.555 | 0.446 | 0.460 | 0.000 | |
| UNSTRACT | 263 | 0.709 | 0.821 | 0.137 | 0.743 | 0.647 | 0.633 | 0.650 | |
| URLLIB3 | 129 | 0.702 | 0.812 | 0.136 | 0.761 | 0.621 | 0.535 | 0.515 | 1.000 |
| VLLM | 858 | 0.326 | 0.634 | 0.485 | 0.337 | 0.313 | 0.273 | 0.186 | 0.175 |
| VNPY | 150 | 0.827 | 0.924 | 0.105 | 0.828 | 0.788 | 0.939 | 0.375 | |

*Table 17.* TYPECHECK and TYPESIM scores for QWEN-2.5-7B on each repository.

| REPO | TYPE CHECK | TYPE SIM | TYPESIM WO MISSING | MISSING RATIO | TYPESIM BY DEPTH | | | | |
|---|---|---|---|---|---|---|---|---|---|
| | | | | | DEPTH1 | DEPTH2 | DEPTH3 | DEPTH4 | DEPTH5 |
| AGENTS | 158 | 0.695 | 0.865 | 0.196 | 0.706 | 0.683 | 0.507 | 0.000 | |
| APPWORLD | N/A | 0.462 | 0.921 | 0.498 | 0.502 | 0.380 | 0.350 | 0.368 | 0.000 |
| AUTOGPT | 252 | 0.680 | 0.784 | 0.132 | 0.703 | 0.633 | 0.596 | 0.626 | 0.750 |
| BLACK | 101 | 0.491 | 0.651 | 0.247 | 0.543 | 0.292 | 0.411 | 0.325 | |
| CAPA | 511 | 0.576 | 0.757 | 0.240 | 0.589 | 0.402 | 0.656 | 0.793 | |
| COMPOSIO | 108 | 0.546 | 0.655 | 0.167 | 0.639 | 0.358 | 0.211 | 0.636 | |
| DB-GPT | 2245 | 0.596 | 0.693 | 0.141 | 0.702 | 0.460 | 0.440 | 0.508 | 0.481 |
| DEEPFACE | 36 | 0.492 | 0.840 | 0.415 | 0.528 | 0.582 | 0.214 | 0.323 | 0.000 |
| EXO | 176 | 0.682 | 0.881 | 0.226 | 0.669 | 0.713 | 0.747 | 0.708 | 0.714 |
| FACESWAP | 89 | 0.643 | 0.844 | 0.238 | 0.748 | 0.329 | 0.151 | 0.229 | 0.380 |
| FASTAPI | 863 | 0.191 | 0.484 | 0.605 | 0.315 | 0.118 | 0.155 | 0.065 | 0.690 |
| FLAKE8 | 35 | 0.662 | 0.887 | 0.253 | 0.715 | 0.546 | 0.571 | 0.750 | |
| FLASK | 116 | 0.715 | 0.836 | 0.145 | 0.797 | 0.560 | 0.400 | 0.167 | |
| GHUNT | 67 | 0.696 | 0.851 | 0.183 | 0.701 | 0.613 | 0.959 | 0.998 | |
| GPT-PILOT | 489 | 0.776 | 0.872 | 0.110 | 0.778 | 0.805 | 0.623 | 0.739 | |
| GPT4FREE | 646 | 0.795 | 0.900 | 0.116 | 0.821 | 0.702 | 0.701 | 0.500 | |
| GPTME | 105 | 0.652 | 0.847 | 0.231 | 0.680 | 0.583 | 0.529 | | 1.000 |
| GRADIO | 2008 | 0.459 | 0.613 | 0.251 | 0.580 | 0.394 | 0.248 | 0.260 | 0.177 |
| HAYSTACK | 330 | 0.326 | 0.528 | 0.383 | 0.493 | 0.136 | 0.078 | 0.296 | |
| LEROBOT | N/A | 0.438 | 0.752 | 0.417 | 0.581 | 0.267 | 0.283 | 0.289 | 0.000 |
| LLAMA-FACTORY | 251 | 0.608 | 0.789 | 0.230 | 0.644 | 0.555 | 0.537 | 0.226 | 0.000 |
| MANIM | N/A | 0.732 | 0.877 | 0.166 | 0.806 | 0.522 | 0.319 | 0.778 | 0.000 |
| MLC-LLM | 114 | 0.527 | 0.777 | 0.322 | 0.585 | 0.388 | 0.232 | 0.358 | 0.137 |
| NICEGUI | 640 | 0.790 | 0.882 | 0.105 | 0.811 | 0.733 | 0.644 | 0.712 | 0.325 |
| OPENAI-PYTHON | 218 | 0.544 | 0.654 | 0.168 | 0.617 | 0.462 | 0.319 | 0.187 | |
| OPENBB | 294 | 0.409 | 0.453 | 0.097 | 0.776 | 0.260 | 0.352 | 0.359 | 0.346 |
| PANDAS-AI | 462 | 0.721 | 0.793 | 0.090 | 0.720 | 0.791 | 0.518 | 0.719 | 1.000 |
| PAPER-QA | 67 | 0.583 | 0.776 | 0.249 | 0.588 | 0.597 | 0.490 | 0.704 | 0.833 |
| PDM | 603 | 0.723 | 0.891 | 0.188 | 0.776 | 0.580 | 0.566 | 0.765 | |
| PHIDATA | 1713 | 0.804 | 0.894 | 0.100 | 0.856 | 0.741 | 0.671 | 0.850 | 0.361 |
| PILLOW | 100 | 0.574 | 0.841 | 0.318 | 0.634 | 0.402 | 0.300 | 0.390 | 0.417 |
| PIP | 1143 | 0.717 | 0.902 | 0.206 | 0.761 | 0.615 | 0.550 | 0.166 | 0.804 |
| POETRY | 132 | 0.854 | 0.932 | 0.084 | 0.903 | 0.708 | 0.619 | 0.676 | 0.958 |
| PRE-COMMIT | 83 | 0.854 | 0.916 | 0.068 | 0.869 | 0.828 | 0.752 | | |
| PRE-COMMIT-HOOKS | 8 | 0.921 | 0.958 | 0.039 | 0.927 | 0.890 | 0.934 | | |
| PRIVATE-GPT | 45 | 0.485 | 0.873 | 0.444 | 0.459 | 0.534 | 0.546 | | |
| PYDANTIC | 606 | 0.383 | 0.789 | 0.514 | 0.445 | 0.269 | 0.201 | 0.111 | 0.000 |
| RICH | 255 | 0.579 | 0.834 | 0.305 | 0.587 | 0.545 | 0.667 | 0.750 | |
| SCREENSHOT-TO-CODE | 14 | 0.756 | 0.918 | 0.177 | 0.837 | 0.683 | 0.616 | 0.833 | |
| SPHINX | 421 | 0.373 | 0.669 | 0.443 | 0.377 | 0.373 | 0.255 | 0.274 | 0.393 |
| SPOTIFY-DOWNLOADER | 31 | 0.429 | 0.637 | 0.327 | 0.497 | 0.286 | 0.386 | 0.167 | |
| STREAMLIT | 195 | 0.287 | 0.426 | 0.326 | 0.246 | 0.528 | 0.882 | | |
| SUPERVISION | 288 | 0.554 | 0.827 | 0.331 | 0.645 | 0.392 | 0.562 | 0.612 | 0.315 |
| TAIPY | 853 | 0.487 | 0.704 | 0.308 | 0.600 | 0.328 | 0.225 | 0.046 | 0.000 |
| TEN-AGENT | 72 | 0.686 | 0.808 | 0.152 | 0.683 | 0.726 | 0.532 | 0.472 | |
| TYPER | 328 | 0.472 | 0.891 | 0.470 | 0.493 | 0.454 | 0.417 | 0.000 | |
| UNSTRACT | 216 | 0.796 | 0.859 | 0.073 | 0.812 | 0.767 | 0.779 | 0.650 | |
| URLLIB3 | 118 | 0.662 | 0.794 | 0.167 | 0.755 | 0.524 | 0.522 | 0.231 | 1.000 |
| VLLM | 1631 | 0.513 | 0.673 | 0.238 | 0.569 | 0.400 | 0.389 | 0.299 | 0.750 |
| VNPY | 82 | 0.816 | 0.841 | 0.030 | 0.857 | 0.808 | 0.140 | 0.000 | |

## D.3. Type Inference Examples

We a code snippet from the repo `DB-GPT` to demonstrate the type inference accuracy of each LLM. We have conducted necessary code formatting for better visualization.

### SOURCE CODE

```python
class RerankEmbeddings(ABC):
    @abstractmethod
    def predict(self, query: str, candidates: List[str]) -> List[float]:
        ...

    async def apredict(self, query: str, candidates: List[str]) -> List[float]:
        ...

class Embeddings(ABC):
    @abstractmethod
    def embed_documents(self, texts: List[str]) -> List[List[float]]:
        ...

    @abstractmethod
    def embed_query(self, text: str) -> List[float]:
        ...

    async def aembed_documents(self, texts: List[str]) -> List[List[float]]:
        ...

    async def aembed_query(self, text: str) -> List[float]:
        ...
```

### GPT-4O (SCORE: 0.874)

```python
class RerankEmbeddings(ABC):
    @abstractmethod
    def predict(self, query: Any, candidates: List[Any]) -> List[float]:
        ...

    async def apredict(self, query: Any, candidates: List[Any]) -> List[float]:
        ...

class Embeddings(ABC):
    @abstractmethod
    def embed_documents(self, texts: List[str]) -> List[List[float]]:
        ...

    @abstractmethod
    def embed_query(self, text: str) -> List[float]:
        ...

    async def aembed_documents(self, texts: List[str]) -> List[List[float]]:
        ...

    async def aembed_query(self, text: str) -> List[float]:
        ...
```

GPT-4O-MINI (SCORE: 0.886)

```
1   class RerankEmbeddings(ABC):
2       @abstractmethod
3       def predict(self, query: Any, candidates: List[Any]) -> List[float]:
4           ...
5
6       async def apredict(self, query: Any, candidates: List[Any]) -> List[float]:
7           ...
8
9   class Embeddings(ABC):
10      @abstractmethod
11      def embed_documents(self, texts: List[str]) -> List[List[float]]:
12          ...
13
14      @abstractmethod
15      def embed_query(self, text: str) -> List[float]:
16          ...
17
18      async def aembed_documents(self, texts: List[str]) -> List[List[float]]:
19          ...
20
21      async def aembed_query(self, text: str) -> List[float]:
22          ...
```

CLAUDE-3.5-SONNET (SCORE: 0.870)

```
1   class RerankEmbeddings(ABC):
2       @abstractmethod
3       def predict(self, query: Any, candidates: List[Any]) -> List[float]:
4           ...
5
6       async def apredict(self, query: Any, candidates: List[Any]) -> List[float]:
7           ...
8
9   class Embeddings(ABC):
10      @abstractmethod
11      def embed_documents(self, texts: List[str]) -> List[List[float]]:
12          ...
13
14      @abstractmethod
15      def embed_query(self, text: str) -> List[float]:
16          ...
17
18      async def aembed_documents(self, texts: List[str]) -> List[List[float]]:
19          ...
20
21      async def aembed_query(self, text: str) -> List[float]:
22          ...
```

DEEPSEEK-V3 (SCORE: 0.898)

```
1   class RerankEmbeddings(ABC):
2       @abstractmethod
3       def predict(self, query: str, candidates: List[str]) -> List[float]:
4           ...
5
6       async def apredict(self, query: str, candidates: List[str]) -> Awaitable[List[float]]:
7           ...
8
9
10  class Embeddings(ABC):
11      @abstractmethod
12      def embed_documents(self, texts: List[str]) -> List[List[float]]:
13          ...
14
15      @abstractmethod
16      def embed_query(self, text: str) -> List[float]:
17          ...
18
19      async def aembed_documents(self, texts: List[str]) -> Awaitable[List[List[float]]]:
20          ...
21
22      async def aembed_query(self, text: str) -> Awaitable[List[float]]:
23          ...
```

DEEPSEEK-V2.5 (SCORE: 0.899)

```
1   class RerankEmbeddings(ABC):
2       @abstractmethod
3       def predict(self, query: str, candidates: List[str]) -> List[float]:
4           ...
5
6       async def apredict(self, query: str, candidates: List[str]) -> List[float]:
7           ...
8
9
10  class Embeddings(ABC):
11      @abstractmethod
12      def embed_documents(self, texts: List[str]) -> List[List[float]]:
13          ...
14
15      @abstractmethod
16      def embed_query(self, text: str) -> List[float]:
17          ...
18
19      async def aembed_documents(self, texts: List[str]) -> List[List[float]]:
20          ...
21
22      async def aembed_query(self, text: str) -> List[float]:
23          ...
```

GROK-2 (SCORE: 0.898)

```
1   class RerankEmbeddings(ABC):
2       @abstractmethod
3       def predict(self, query: str, candidates: List[str]) -> List[float]:
4           ...
5
6       def apredict(self, query: str, candidates: List[str]) -> asyncio.Future[List[float]]:
7           ...
8
9   class Embeddings(ABC):
10      @abstractmethod
11      def embed_documents(self, texts: List[str]) -> List[List[float]]:
12          ...
13
14      @abstractmethod
15      def embed_query(self, text: str) -> List[float]:
16          ...
17
18      def aembed_documents(self, texts: List[str]) -> asyncio.Future[List[List[float]]]:
19          ...
20
21      def aembed_query(self, text: str) -> asyncio.Future[List[float]]:
22          ...
```

LLAMA-3-8B (SCORE: 0.604)

```
1   class RerankEmbeddings(ABC):
2       @abstractmethod
3       def predict(self, query: object, candidates: List[object]) -> object:
4           ...
5
6       async def apredict(self, query: object, candidates: List[object]) -> Awaitable[object]:
7           ...
8
9   class Embeddings(ABC):
10      @abstractmethod
11      def embed_documents(self, texts: List[object]) -> List[object]:
12          ...
13
14      @abstractmethod
15      def embed_query(self, text: object) -> object:
16          ...
17
18      async def aembed_documents(self, texts: List[object]) -> Awaitable[List[object]]:
19          ...
20
21      async def aembed_query(self, text: object) -> Awaitable[object]:
22          ...
```

LLAMA-3.1-8B (SCORE: 0.769)

```python
class RerankEmbeddings(ABC):
    @abstractmethod
    def predict(self, query: Any, candidates: List[Any]) -> Any:
        ...

    async def apredict(self, query: Any, candidates: List[Any]) -> Awaitable[Any]:
        ...

class Embeddings(ABC):
    @abstractmethod
    def embed_documents(self, texts: List[Any]) -> List[Any]:
        ...

    @abstractmethod
    def embed_query(self, text: Any) -> Any:
        ...

    async def aembed_documents(self, texts: List[Any]) -> Awaitable[List[Any]]:
        ...

    async def aembed_query(self, text: Any) -> Awaitable[Any]:
        ...
```

QWEN-2.5-7B (SCORE: 0.693)

```python
class RerankEmbeddings(ABC):
    @abstractmethod
    def predict(self, query: str, candidates: List[str]) -> List[float]:
        ...

    async def apredict(self, query: str, candidates: List[str]) -> List[float]:
        ...

class Embeddings(ABC):
    @abstractmethod
    def embed_documents(self, texts: List[str]) -> List[List[float]]:
        ...

    @abstractmethod
    def embed_query(self, text: str) -> List[float]:
        ...

    async def aembed_documents(self, texts: List[str]) -> List[List[float]]:
        ...

    async def aembed_query(self, text: str) -> List[float]:
        ...
```

