# OpenReview forum: "TypyBench: Evaluating LLM Type Inference for Untyped Python Repositories"
_ICML.cc/2025/Conference — ICML 2025 poster_

### Official Review · Reviewer_KRyD · 2025-03-04

**Overall Recommendation:** 3

**Summary:**

The paper introduces TYPYBENCH, a benchmark for evaluating the capability of Large Language Models (LLMs) in type inference for Python repositories without explicit type annotations. It defines two novel metrics: TYPESIM, which captures semantic relationships between predicted and ground truth types using syntax-level features, and TYPECHECK, which assesses type consistency across entire codebases. The benchmark includes 50 well-typed Python repositories from diverse domains. Experimental results show that while LLMs achieve reasonable type similarity, they struggle with complex nested types.

**Claims And Evidence:**

Yes.

**Essential References Not Discussed:**

N/A.

**Experimental Designs Or Analyses:**

I checked how TYPESIM works in detail and appreciate the authors’ efforts to move beyond exact type matching by incorporating semantic similarity.
However, while TYPESIM aims to capture the semantic relationships between types, it still primarily relies on syntax-level comparisons and focuses on shared fields and attributes between types.
This raises concerns about whether it fully captures deeper semantic equivalences beyond structural similarity.

Additionally, the study acknowledges that LLMs are evaluated file by file due to token size limitations.
It raises a concern that the capability of LLMs in type inference could be undermined due to the limited size of context windows.
While I acknowledge that the experimental results are persuasive, it would be great if the authors could discuss using techniques like using techniques as program slicing to feed related context of a type to LLMs.

**Methods And Evaluation Criteria:**

Yes.

**Other Comments Or Suggestions:**

N/A.

**Other Strengths And Weaknesses:**

### Strengths

- The type analysis is performed across both built-in, user-defined, and generic types.

- The benchmark is targeted at the type inference on the repository level, across 50 well-typed Python repositories from diverse domains.

- The evaluation is comprehensive, including multiple state-of-the-art LLMs.


### Weaknesses

- TYPESIM still relies on syntax-level comparisons.
- Lack of comparison between baseline static type inference tools but involving mypy only.

**Questions For Authors:**

- LLMs are evaluated file by file. Can the authors discuss the potential limitations of using a single file for type inference and possible future directions in improving LLM's type inference via giving more constructive contexts?

- On the lack of comparison with static type inference tools. From my knowledge, there are other tools than Mypy such as Pyre developed by Meta are doing Python's type analysis.  While I am not suggesting additional experiments, could the authors discuss the limitations of relying solely on Mypy as the baseline? How could alternative static analysis tools influence the evaluation results?

- TYPESIM only relies on the similarity of syntax features. While TYPESIM effectively relaxes exact type matching, it primarily relies on syntax-level features, such as attribute comparisons. Have you considered incorporating other features related to code semantics? For example, the similarity between the control flow graphs of two functions.

**Relation To Broader Scientific Literature:**

The paper mainly introduces a benchmark for Python's type analysis on the repository level, which could highlight future research in studying the how LLMs can help type inference of industry length.

**Theoretical Claims:**

Not applicable, the paper is about benchmarks and there are no theoretical claims or proofs.

---

> ### Author Rebuttal · Authors · 2025-04-01
>
> Thanks for your valuable feedback and suggestions. We would like to address your remaining concerns with the following responses. We will improve the manuscript accordingly to address them.
>
> > LLMs are evaluated file by file. Can the authors discuss the potential limitations of using a single file for type inference and possible future directions in improving LLM's type inference via giving more constructive contexts?
>
> Thanks for the suggestion. We did a preliminary case study and found that a common cause of type inconsistency comes from missing context (defined in other files or imported from third-party libraries). The patterns include but not limited to:
>
> 1. Missing base class
> ```python
> class A(B):
>     def func(vars):
>         super().func(vars)
> ```
>
> 2. Missing function information
> ```python
> def func(vars):
>     return another_func(vars)
> ```
>
> 3. Missing decorator information
> ```python
> @some_decorator
> def func(vars):
>     …
> ```
>
> The most straightforward way is to provide the whole repository as the context, but limited by LLM's context window and might hurt TypeSim as shown in Table 4.
>
> An alternative way is to copy-paste the necessary context following the `import` statements but requires engineering efforts. We tried to provide such context manually for the examples we found, and the inconsistency issues were resolved after the context was provided.
>
> Another potential method could be iteratively refining the stub files while using all stub files (and the implementation of the current file) as context. This would require much less context window since the stub files are relatively small.
>
> > On the lack of comparison with static type inference tools. From my knowledge, there are other tools than Mypy such as Pyre developed by Meta are doing Python's type analysis. While I am not suggesting additional experiments, could the authors discuss the limitations of relying solely on Mypy as the baseline? How could alternative static analysis tools influence the evaluation results?
>
> Thanks for bringing up this point. While Mypy is imperfect and could produce false positives or incorrect errors, it is still a popular choice for software engineering best practices to improve code quality. Evaluation results may be different when switching to another static type inference tool like Pyre, but since the majority of static checks should be similar, the change in the evaluation results should be small. TypeCheck metric could switch to another static type inference tool if there is a better and well-adopted one.
>
>
> > TYPESIM only relies on the similarity of syntax features. While TYPESIM effectively relaxes exact type matching, it primarily relies on syntax-level features, such as attribute comparisons. Have you considered incorporating other features related to code semantics? For example, the similarity between the control flow graphs of two functions.
>
> Thanks for your question. If our understanding is correct, you are suggesting not only comparing whether two types have the same functions, but also examining the code semantics, like how similar are their implementations for the same function?
>
> We think it is an interesting direction for future work (especially for complex user-defined types), but current TypeSim already reasonably captures the similarity between types with convenient and robust calculations.
>
> | Original | Predicted | TypeSim  |
> |-|-|-|
> | `list[Any] \| None` | `list[str] \| None`| 0.75|
> | `dict[Any, Any] \| None` | `dict[str, Any] \| None`| 0.875 |
> | `dict[str, tuple[Any, ...]] \| None` | `dict[str, tuple[int, ...]] \| None`| 0.9375|
> | `dict[str, list[int]]`| `dict[str, Union[tuple[int, ...], Any]]`| 0.8385|
> | `float \| np.ndarray[Any, Any]`| `np.ndarray[Any, Any]`| 0.5|
> | `pathlib.Path \| None`| `str \| pathlib.Path \| None`| 0.6667 |
>
> Above is a list of type pairs and their TypeSim score. It can be seen that the TypeSim score reflects the compatibility of two types, which roughly matches human's impression on the type similarity.

---

### Official Review · Reviewer_MFpm · 2025-03-14

**Overall Recommendation:** 4

**Summary:**

In this work, the authors evaluate the ability of LLMs to perform type inference in Python codebases. They introduce two type inference evaluation metrics: (1) TypeSim, which extends prior work focused on exact matching to consider semantic similarity between LLM-inferred vs. human-annotated types, and (2) TypeCheck, which assess repository-level type consistency (e.g., detection of incompatible return types and invalid argument types). They also introduce a type inference benchmark dataset, TypyBench, which consists of a curated set of 50 Python repos. Their empirical results indicate that LLMs are better at local type inference (as indicated by TypeSim scores) than global consistency (as indicated by the TypeCheck scores).

**Claims And Evidence:**

The authors' empirical results largely support their claims and validate the benefits of the metrics they propose.

**Essential References Not Discussed:**

I am not aware of any essential but omitted references.

**Experimental Designs Or Analyses:**

I read their empirical evaluation section, but did not take further steps to verify/replicate the soundness of their results.

**Methods And Evaluation Criteria:**

The metrics that the authors introduce (e.g., TypeSim and TypeCheck) and the ways in which they operationalize and evaluate them seem appropriate and useful.

**Other Comments Or Suggestions:**

- It might be helpful to provide technical definitions for consistency and coherence when these terms are used early in the paper.
- If possible, it would be helpful to place algorithm blocks after the text that describes them, rather than before.

**Other Strengths And Weaknesses:**

Strengths
-  The paper is well-written and well-motivated.
- Consideration of the semantic similarity of type annotations meaningfully improves upon evaluation based on exact matching.
- The authors conduct empirical analysis on a representative set of LLMs and a well-curated benchmark dataset.

Weaknesses
- It would be helpful to provide error analysis/insights into how/why LLMs struggle with code consistency. This can help to inform mitigation efforts that do not require passing the whole repo in as context.

**Questions For Authors:**

1. Have you considered alternative methods for assessing type/argument similarity? (eg, embedding-based approaches?)

**Relation To Broader Scientific Literature:**

This paper is related to prior work on type inference methods, including static/dynamic analysis, ML-based approaches that leverage structured and/or unstructured representations of code, and existing type inference and programming-related benchmark datasets (e.g., for code generation, code completion, debugging, etc.).

**Theoretical Claims:**

N/A, the authors’ claims are largely empirical.

---

> ### Author Rebuttal · Authors · 2025-04-01
>
> Thanks for your valuable feedback and suggestions. We would like to address your remaining concerns with the following responses. We will improve the manuscript accordingly to address them.
>
> > Have you considered alternative methods for assessing type/argument similarity? (eg, embedding-based approaches?)
>
> Our attribute-based method (L198) can also be viewed as an embedding-based approach.
> For example, if we regard each attribute as a column (1 means has and 0 means not), then each type gets an embedding representation, then TypeSim is naturally defined as the Jaccard Similarity over such a representation.
>
> For example:
>
> Type | __iter__ | append | __add__ | … |
> |-|-|-|-|-|
> int | 0 | 0 | 1| …|
> list | 1 | 1 | 1| …|
> Sequence | 1 | 0 | 0| …|
>
> Such embedding representation can naturally work for user-defined types and the similarity is based on the functionality of types rather than the semantic meaning of the name of the types.
>
> > It would be helpful to provide error analysis/insights into how/why LLMs struggle with code consistency. This can help to inform mitigation efforts that do not require passing the whole repo in as context.
>
> Thanks for the suggestion. We did a preliminary case study and found that a common cause of type inconsistency comes from missing context (defined in other files or imported from third-party libraries). The patterns include but not limited to:
>
> 1. Missing base class
> ```python
> class A(B):
>     def func(vars):
>         super().func(vars)
> ```
>
> 2. Missing function information
> ```python
> def func(vars):
>     return another_func(vars)
> ```
>
> 3. Missing decorator information
> ```python
> @some_decorator
> def func(vars):
>     …
> ```
>
> The most straightforward way is to provide the whole repository as the context, but limited by LLM's context window and might hurt TypeSim as shown in Table 4.
>
> An alternative way is to copy-paste the necessary context following the `import` statements but requires engineering efforts. We tried to provide such context manually for the examples we found, and the inconsistency issues were resolved after the context was provided.
>
> Another potential method could be iteratively refining the stub files while using all stub files (and the implementation of the current file) as context. This would require much less context window since the stub files are relatively small.
>
> > It might be helpful to provide technical definitions for consistency and coherence when these terms are used early in the paper.
>
> > If possible, it would be helpful to place algorithm blocks after the text that describes them, rather than before.
>
> Thanks for both suggestions. We will improve the manuscript accordingly.

---

### Official Review · Reviewer_zv9m · 2025-03-14

**Overall Recommendation:** 3

**Summary:**

The paper introduces TypyBench, a benchmark aimed at evaluating large language models (LLMs) on their ability to perform type inference for Python code. Recognizing limitations in existing benchmarks and exact-matching metrics, the authors propose two novel evaluation measures: TypeSim, which assesses semantic similarity between predicted and ground truth types, and TypeCheck, which evaluates type consistency across repositories using static type checking. Through experiments on a curated dataset of 50 Python repositories, the study reveals that while state-of-the-art LLMs can achieve strong semantic similarity scores, they struggle with maintaining consistent type annotations at a repository scale, especially for complex or nested types.

## update after rebuttal
I appreciate the authors' responses, which conceptually addressed my concerns, though open-sourcing the whole benchmark at this moment will be more appreciated than 1-2 examples in the rebuttal to convince me regarding the benchmark design and quality.

I am raising my score to support this paper's acceptance for now, though I am still not fully convinced without seeing the real examples in the benchmark.

**Claims And Evidence:**

Claims in this paper are mostly supported by empirical evidence.

**Essential References Not Discussed:**

References are well discussed.

**Experimental Designs Or Analyses:**

See "Methods And Evaluation Criteria".

**Methods And Evaluation Criteria:**

__Need to Justify Why We Need LLMs to Do Type Inference__

While this paper presents a well-structured benchmark and introduces valuable evaluation metrics (TypeSim and TypeCheck) for assessing LLMs' performance on type inference, it lacks a compelling justification for the necessity of using LLMs in this context. Given the existence of mature and widely used symbolic tools such as Mypy, Pyright, and MonkeyType, the paper should more clearly articulate what specific limitations of symbolic approaches LLMs are intended to address. Without a thorough discussion of the practical advantages of LLMs—such as their ability to generalize to uncommon patterns, handle incomplete code, or infer types in low-resource settings—it remains unclear why LLM-based inference is worth pursuing over traditional methods.

To strengthen the contribution, I encourage the authors to (1) provide a clear motivation for using LLMs for type inference, including a comparison of their theoretical or practical advantages over symbolic systems, and (2) include an empirical baseline where symbolic tools are evaluated on the same benchmark. Demonstrating that LLMs can outperform symbolic methods—either in terms of TypeSim, TypeCheck, or overall utility in real-world development scenarios—would substantially reinforce the paper’s significance and help clarify the role LLMs should play in the type inference landscape.

__Usefulness of Applying TypyBench as a Mainstream Evaluation Benchmark__

Another key concern lies in the broader motivation for evaluating LLMs specifically on the task of type inference. As a benchmark, TypyBench would benefit from a clearer articulation of what general model capabilities type inference serves as a proxy for. Unlike tasks such as code generation or execution reasoning—where LLMs are uniquely positioned to outperform symbolic methods and improve downstream applications like self-debugging or automatic code generation —type inference is a narrow and well-defined task that symbolic tools already handle effectively in many practical settings. Without a strong argument for how improving LLMs' performance on type inference translates into broader gains in language model capabilities, it remains unclear why this task warrants dedicated benchmarking and why LLM developers should prioritize it as part of their evaluation pipeline.

To enhance the benchmark’s relevance, the authors should consider framing type inference within a more holistic view of code understanding or software engineering tasks that LLMs uniquely enable. Alternatively, they could provide evidence that performance on type inference correlates with performance on broader tasks like code editing, completion, or static analysis augmentation—thereby justifying the benchmark as a meaningful diagnostic tool. Without this contextualization, the current scope risks appearing too narrow and decoupled from the more impactful capabilities that developers and practitioners typically seek in modern LLMs.

__More Illustration Regarding the New Metrics__

While the proposed TypeSim and TypeCheck metrics are novel and address some limitations of exact match evaluation, their design and practical significance could benefit from further clarification. TypeSim, in particular, introduces a similarity-based measure that may capture more semantic nuance, but it’s not entirely clear how well it aligns with human judgment of correctness or practical utility in real-world development workflows. Similarly, TypeCheck measures consistency via mypy errors, but it would be helpful to better justify why this is a comprehensive proxy for type quality. I encourage the authors to provide more rationale or empirical validation for these metrics—perhaps through user studies, ablation analysis, or correlation with downstream developer effort—to strengthen confidence in their effectiveness and generalizability.

**Other Comments Or Suggestions:**

See "Methods And Evaluation Criteria".

**Other Strengths And Weaknesses:**

See "Methods And Evaluation Criteria".

**Questions For Authors:**

See "Methods And Evaluation Criteria".

**Relation To Broader Scientific Literature:**

This is related to program analysis literature in PL/SE domain.

**Theoretical Claims:**

N/A

---

> ### Author Rebuttal · Authors · 2025-04-01
>
> Thanks for your valuable feedback and suggestions. We would like to address your remaining concerns with the following responses. We will improve the manuscript accordingly to address them.
>
> ### Need to Justify Why We Need LLMs to Do Type Inference
>
> As shown in [previous work](https://github.com/secure-software-engineering/TypeEvalPy) [1], LLMs already showed better type inference results compared with symbolic tools such as Pyright and [Jedi](http://jedi.readthedocs.io/). Moreover, code-completion tools like Github Copilot and Cursor are making progressively better type hints completions in Python. These motivated us to mainly focus on using LLMs for type inference.
>
> > the paper should more clearly articulate what specific limitations of symbolic approaches LLMs are intended to address
>
> Thanks for the suggestion. We use Jedi to illustrate the limitation of symbolic approaches.
>
> The original code snipet in the `black` repo is:
> ```python
> @dataclass
> class BracketTracker:
>     """Keeps track of brackets on a line."""
>
>     depth: int = 0
>     bracket_match: dict[tuple[Depth, NodeType], Leaf] = field(default_factory=dict)
>     delimiters: dict[LeafID, Priority] = field(default_factory=dict)
>     previous: Optional[Leaf] = None
>     _for_loop_depths: list[int] = field(default_factory=list)
>     _lambda_argument_depths: list[int] = field(default_factory=list)
>     invisible: list[Leaf] = field(default_factory=list)
> ```
>
> The predicted types by *jedi* are:
> ```python
> @dataclass
> class BracketTracker:
>     depth: int
>     bracket_match: dict
>     delimiters: dict
>     previous: None
>     _for_loop_depths: list
>     _lambda_argument_depths: list
>     invisible: list
> ```
> The TypeSim score on `depth` is 1 and 0.5 for others. The overall score for this file is 0.6645.
>
> the predicted types by claude-3.5.sonnet.
> ```python
> class BracketTracker:
>     depth: int
>     bracket_match: Dict[tuple[int, int], Leaf]
>     delimiters: Dict[int, int]
>     previous: Optional[Leaf]
>     _for_loop_depths: List[int]
>     _lambda_argument_depths: List[int]
>     invisible: List[Leaf]
> ```
> The TypeSim scores for these variables are all 1.0 and the overall score of the file is 0.8906.
>
> As shown in the example above, the type predicted by Jedi lost fine-grained information, while Claude recovered most of that with a much higher similarity compared with the original codes.
>
> ### Usefulness of Applying TypyBench as a Mainstream Evaluation Benchmark
>
> > what general model capabilities type inference serves as a proxy for
>
> Thanks for the suggestion. TypyBench mainly introduces the following challenges (see Table 1 for concrete numbers):
>
> 1. Code understanding in long input context length (repo-level, could be >1M tokens)
> 2. Long completion length (each function argument and return value need to be typed)
> 3. Consistency among lots of outputs (i.e., cases, the predicted types)
>
> As shown in Figure 4 and Table 2, the relative performance of tested LLMs (claude-3.5, gpt-4o, grok2, and deepseek-v3 were top-tier at that time) generally matches the relative performance on other coding tasks like the code completion in [LiveCodeBench](https://livecodebench.github.io/leaderboard.html) (claude-3.5: 32, gpt-4o: 30, gpt-4o-mini: 27.7, date range 8/1/2024 - 2/1/2025), indicating it is a meaningful diagnostic benchmark that pressure tests the code understanding ability of long-context LLMs.
>
> ### More Illustration Regarding the New Metrics
>
> > it’s not entirely clear how well it aligns with human judgment of correctness or practical utility in real-world development workflows.
>
> | Original | Predicted | TypeSim  |
> |-|-|-|
> | `list[Any] \| None` | `list[str] \| None`| 0.75|
> | `dict[Any, Any] \| None` | `dict[str, Any] \| None`| 0.875 |
> | `dict[str, tuple[Any, ...]] \| None` | `dict[str, tuple[int, ...]] \| None`| 0.9375|
> | `dict[str, list[int]]`| `dict[str, Union[tuple[int, ...], Any]]`| 0.8385|
> | `float \| np.ndarray[Any, Any]`| `np.ndarray[Any, Any]`| 0.5|
> | `pathlib.Path \| None`| `str \| pathlib.Path \| None`| 0.6667 |
>
> Above is a list of type pairs that are similar but not exactly the same. It can be seen that the TypeSim score reasonably exhibits the similarity between the two types.
>
> > it would be helpful to better justify why this is a comprehensive proxy for type quality.
>
> Mypy is a widely adopted best practice in software engineering to improve code quality and detect potential bugs with static checks. For many well-maintained Python open-source repositories, the number of Mypy errors should be almost 0. Therefore, the number of Mypy errors reflects the developer's efforts to fix the inconsistency after using type inference methods.
>
> [1] Shivarpatna Venkatesh, Ashwin Prasad, et al. "Typeevalpy: A micro-benchmarking framework for python type inference tools." Proceedings of the 2024 IEEE/ACM 46th International Conference on Software Engineering: Companion Proceedings. 2024.

---

### Official Review · Reviewer_SnQQ · 2025-03-15

**Overall Recommendation:** 2

**Summary:**

Summary: The paper introduces TYPYBENCH, a benchmark designed to evaluate the type inference capabilities of large language models (LLMs) across entire Python repositories. The benchmark features two novel metrics: TYPESIM, which measures the semantic similarity between predicted and ground truth types, and TYPECHECK, which evaluates type consistency across codebases. The authors evaluate various LLMs on a dataset of 50 high-quality Python repositories and find that while LLMs achieve reasonable TYPESIM scores, they struggle with complex nested types and exhibit significant type consistency errors. The findings suggest that future research should prioritize improving repository-level type consistency over type similarity. TYPYBENCH provides a foundation for this new research direction, offering insights into model performance across different type complexities and usage contexts.

**Claims And Evidence:**

This paper presents clear and compelling evidence. For instance, Section 2.1 cites relevant references to conventional methods, learning-based methods, and LLM-based methods.

**Essential References Not Discussed:**

The related works are sufficiently discussed in this paper.

**Experimental Designs Or Analyses:**

The experimental design appears reasonable, as it includes both the Main Evaluation Analysis and the Factors Analysis.

**Methods And Evaluation Criteria:**

As far as I know, there are currently no approaches that rely entirely on large language models for type inference, primarily due to concerns around soundness. Given this, why do we need evaluation metrics that are specifically designed for LLMs?
For TypeSim, which aims to evaluate the semantic similarity between predicted and ground-truth types — how frequently does such semantic similarity actually occur at the function level in real-world code? In other words, what proportion of function-level type annotations can be considered semantically interchangeable?
For TypeCheck, which evaluates consistency of types within a codebase — what is the proportion of types in real-world repositories that exhibit this kind of internal consistency? Without quantitative insights into these questions, it's hard to assess how broadly useful or necessary these two criteria are.

**Other Comments Or Suggestions:**

None

**Other Strengths And Weaknesses:**

Strengths:
+ The dataset curation in this paper is a valuable contribution. Overall, it is necessary and well-justified, especially considering that the widely-used dataset currently available is ManyTypes4Py.

+ Clear structure. The structure of this paper is clear.

Weaknesses:
- The motivation behind this work is worth discussing. As far as I know, there already exist a wide range of metrics for evaluating type inference, such as exact match, match to parameter, and others as mentioned in your paper. So why do we need a dedicated metric specifically for LLMs? This point is not clearly addressed in the introduction. Moreover, you mention in the introduction that types like List and Sequence can be used interchangeably. However, this comes across as a special-case observation rather than a generalizable insight. Have you conducted any statistical analysis to quantify how often such interchangeable usage occurs in real-world codebases? Moreover, can other type pairs like str and bool also be used interchangeably? If so, how frequent are such cases?

In addition, match to parameter seems to perform better compared to the proposed TypeSim, as it can match the outer type structure more effectively.Can I then understand TypeSim as a suboptimal alternative to match to parameter, rather than a fundamentally better metric?

-The methodology behind the proposed criteria seems overly simplistic. For TypeSim, the base type similarity is computed using the Jaccard index, followed by a basic additive scheme to estimate structural similarity.But is this really how large language models perform type inference — through such naive calculations? This method appears to have little to do with the actual mechanisms or capabilities of LLMs, and does not take into account any of the unique characteristics or behaviors of language models.
This paper uses the number of mypy errors as a measure of Type Consistency, but this approach is rather naive. mypy itself can produce false positives or incorrect error reports, meaning that the evaluation may not accurately reflect true inconsistencies.


- As a benchmark metric, the current selection of foundation models is insufficient. Notably, the evaluation lacks comparisons with more diverse models such as GPT-4, CodeLlama, StarCoder, CodeT5, and its variants. Including these models would provide a more comprehensive and convincing assessment of the proposed metrics.

-I could not find any links to the open-source code or dataset in this paper. Providing access to the implementation and data is essential for reproducibility and for allowing others to validate or build upon this work.

**Questions For Authors:**

1.why do we need a dedicated metric specifically for LLMs?
2.Have you conducted any statistical analysis to quantify how often such interchangeable usage occurs in real-world codebases?
3.Can I then understand TypeSim as a suboptimal alternative to match to parameter, rather than a fundamentally better metric?
4.Is it reasonable to evaluate the type inference capabilities of LLMs using such simple, naive calculations? Are there more suitable evaluation methods that better align with the way LLMs actually reason about code?

**Relation To Broader Scientific Literature:**

This paper proposes two metrics for LLM-based type inference: TypeSim and TypeCheck.

**Theoretical Claims:**

The paper does not provide specific theoretical proofs, as it is not a theoretical paper.

---

> ### Author Rebuttal · Authors · 2025-04-01
>
> Thanks for your valuable feedback and suggestions. We would like to clarify the metrics design and address your remaining concerns with the following responses. We will improve the manuscript accordingly to address them.
>
> > The motivation behind this work is worth discussing. … Why do we need evaluation metrics that are specifically designed for LLMs?
>
> [Q1] First, we want to clarify that our metrics are not only designed for LLMs but all predictions made by any type inference method (including the program-based ones) can be evaluated using these metrics. These metrics are calculated over a set of type predictions.
>
> **For TypeSim**, as stated in the Introduction (L38), exact matching fails to capture important semantic relationships between types.
>
> [Q3] For *match up to parametric type*, it is a special case of our TypeSim metric as formulated in L187:
>
> $S(T, T')=\alpha~s(root, root')+(1-\alpha)S_{list}(args(T),args(T'))$
>
> where in TypeSim $\alpha=0.5$ and $s$ is Jaccard on attributes, and in *match up to parametric type* $\alpha=1$ and $s$ is exact matching (equivalent to taking floor operation on the result of Jaccard Index). Therefore, TypeSim is a more general similarity metric than *exact match* and *match up to parametric type*.
>
> For example, for a target type `list[str]` and the prediction is `list`, exact matching gives a score of 0 would be too harsh, and *match up to parametric type* gives a score of 1 is too benign. TypeSim gives a score of $\alpha=0.5$ reflects the similarity between two types.
>
> [Q2] The main goal of the TypeSim metric is not limited to detecting the types that could be used interchangeably (which are illustrative examples) but also providing a better measurement for the semantic similarity between complex types by considering their functional attributes (Section 4.1.1 L209-219) and the structure (Section 4.1.2) within the types.
>
> As shown in the experiments (Section 6.3.1), TypeSim provides a more nuanced semantic-based evaluation compared with exact matching, especially for more nested types (e.g., depth > 3).
>
> |Original|Predicted |TypeSim|
> |-|-|-|
> |`list[Any] \| None`|`list[str] \| None`|0.75|
> |`dict[Any, Any] \| None`|`dict[str, Any] \| None`|0.875|
> |`dict[str, tuple[Any, ...]] \| None`|`dict[str, tuple[int, ...]] \| None`| 0.9375|
> |`dict[str, list[int]]`|`dict[str, Union[tuple[int, ...], Any]]`|0.8385|
> |`float \| np.ndarray[Any, Any]`|`np.ndarray[Any, Any]`|0.5|
> |`pathlib.Path \| None`|`str \| pathlib.Path \| None`| 0.6667|
>
> Above is a list of similar type pairs. It can be seen that the TypeSim score reasonably exhibits the similarity between the two types.
>
> **For TypeCheck**, though Mypy is not a perfect type checker and could produce false positives or incorrect error reports, it is currently a well-adopted best practice for software engineering. Repositories passing Mypy check (or having fewer Mypy errors) tend to have fewer bugs in practice, so we introduce this as a good starting point for evaluating type inference results. We will discuss this limitation and leave it as future work to find a better checking mechanism.
>
> We hope these clarifications resolve your questions. Please let us know if you have further questions.
>
> > The evaluation lacks comparisons with more diverse models such as GPT-4, CodeLlama, StarCoder, CodeT5, and its variants.
>
> We did not evaluate these LLMs since they are already outdated models compared with the SOTA LLMs we tested (e.g., GPT-4o, Claude-3.5-sonnet). Moreover, SOTA general-purpose models (e.g., GPT-4o) have already surpassed code-focused models in coding tasks. Nonetheless, we tested CodeLlama on the test sets as suggested, which gives similar results as Llama-3-8B:
>
> |Model|TypeCheck|TypeSim|TypeSim wo missing|Missing rate|
> |-|-|-|-|-|
> |Llama-3-8b|44.0|0.396|0.747|0.470|
> |CodeLlama|25.0*|0.3558|0.735|0.516|
>
> *: only count on private-gpt
>
> **Other questions**,
>
> > [Q4] Is it reasonable to evaluate the type inference capabilities of LLMs using such simple, naive calculations? Are there more suitable evaluation methods that better align with the way LLMs actually reason about code?
>
> We’d like to clarify that our proposed metrics — TypeSim and TypeCheck — are not designed to mimic or replicate how LLMs internally reason about code or types. Instead, they are metrics measuring the semantic quality and repository-wide consistency of predicted types (regardless of how those predictions are produced, by LLMs or symbolic tools). TypeSim provides an interpretable view of type prediction quality, especially when predictions are partially correct. Its formulation using Jaccard similarity and structural decomposition is a pragmatic and interpretable approach that aligns with how types are composed and used in practice.
>
> > I could not find any links to the open-source code or dataset in this paper.
>
> We will open-source the code and dataset when published.

---

### Decision · Program_Chairs · 2025-05-01

**Decision:**

Accept (poster)

**Comment:**

**Meta-Review for “TypyBench: Evaluating LLM Type Inference for Untyped Python Repositories”**

---

### Summary of the Paper

The submission introduces **TypyBench**, a new benchmark for evaluating Python type inference—particularly with large language models (LLMs)—on entire code repositories rather than just on isolated files or functions. The authors propose two metrics:
1. **TypeSim**, which measures how semantically similar a predicted type is to the “ground truth” by looking beyond exact string matches (e.g., capturing that `List[str]` and `Sequence[str]` share functional attributes).
2. **TypeCheck**, which evaluates repository-level consistency by counting how many type errors (e.g., incompatible return types, invalid arguments) a static type checker (mypy) reports when using the inferred annotations.

An dataset of 50 curated Python repositories is collected and processed so that all original type annotations are removed (thus forming the benchmark’s input). The repositories are chosen to be sufficiently diverse (e.g., developer tools, ML libraries, security, web APIs) and to have meaningful, correct, and relatively complete annotations in their original form. The authors demonstrate that state-of-the-art LLMs can achieve decent scores on local type similarity (TypeSim) but do not ensure consistency well at the repository scale (TypeCheck), particularly for nested or complex types.

---

### Strengths

1. **Type Inference Metrics**
   - **TypeSim** captures partial correctness. This is an improvement over pure exact-match or “match up to parametric type” metrics that might over-penalize cases where the model predicts, for example, `List` instead of `List[int]`.
   - **TypeCheck** indicates how well an inferred set of types holds together globally, which is important in real-world usage.

2. **Type Inference Dataset**
   - A collection of 50 repositories across diverse domains.
   - The dataset is large enough (over 100k total lines in many repos) to test long-context and consistency challenges.

3. **Evaluation on Multiple Models**
   - The experiments compare both top-tier API-based models (GPT-4 variants, Claude, Deepseek, Grok) and smaller local models (Llama variants, Qwen).
   - Results show that some models excel at local similarity (TypeSim) but suffer from consistency errors (TypeCheck).

---

### Weaknesses

1. **Comparisons to Static Analysis Tools**
   - Comparison with other static-analysis tools would improve the evaluation.

2. **Drawbacks of Proposed Metrics**
   - **TypeSim** uses syntax level comparisons comparing sets of supported methods and structural features. While this is an improvement over exact matching, some reviewers expressed skepticism as to whether it is enough to fully capture “semantic” type correctness.
   - **TypeCheck** is limited by mypy’s correctness. False positives or negatives do exist, although it is considered a fair standard for everyday Python usage.

---

### **Recommendation**:

Based on reviewers evaluations, I recommend weak accept.